# Disruption of day-to-night changes in circadian gene expression with chronic tendinopathy

Ching-Yan Chloé Yeung[1,2] , René B. Svensson[1,2], Kateryna Yurchenko[1,2],
Nikolaj M. Malmgaard-Clausen[1,2], Ida Tryggedsson[1,2], Marius Lendal[1,2] , Anja Jokipii-Utzon[1,2],
Jens L. Olesen[1,2], Yinhui Lu[3], Karl E. Kadler[3], Peter Schjerling[1,2] and Michael Kjær[1,2]

[1]*Institute of Sports Medicine Copenhagen, Department of Orthopedic Surgery, Copenhagen University Hospital – Bispebjerg and Frederiksberg, Copenhagen, Denmark*
[2]*Center for Healthy Aging, Department of Clinical Medicine, University of Copenhagen, Denmark*
[3]*Wellcome Centre for Cell-Matrix Research, Faculty of Biology, Medicine and Health, University of Manchester, Manchester, UK*

Handling Editors: Michael Hogan & Martino Franchi

The peer review history is available in the Supporting information section of this article
(https://doi.org/10.1113/JP284083#support-information-section).

<div style="writing-mode: vertical">The Journal of Physiology</div>

**Abstract**  Overuse injury in tendon tissue (tendinopathy) is a frequent and costly musculoskeletal disorder and represents a major clinical problem with unsolved pathogenesis. Studies in mice have demonstrated that circadian clock-controlled genes are vital for protein homeostasis and important in the development of tendinopathy. We performed RNA sequencing, collagen content and ultrastructural analyses on human tendon biopsies obtained 12 h apart in healthy individuals to establish whether human tendon is a peripheral clock tissue and we performed RNA sequencing

**Chloé Yeung** has a PhD in Cell Biology from the University of Manchester. She is a senior researcher at the Institute of Sports Medicine Copenhagen at Bispebjerg Hospital. Her research focuses on understanding the role of the circadian clock in tendon extracellular matrix homeostasis in health and in disease, for example, tendinopathy.

on patients with chronic tendinopathy to examine the expression of circadian clock genes in tendinopathic tissues. We found time-dependent expression of 280 RNAs including 11 conserved circadian clock genes in healthy tendons and markedly fewer (23) differential RNAs with chronic tendinopathy. Further, the expression of COL1A1 and COL1A2 was reduced at night but was not circadian rhythmic in synchronised human tenocyte cultures. In conclusion, day-to-night changes in gene expression in healthy human patellar tendons indicate a conserved circadian clock as well as the existence of a night reduction in collagen I expression.

(Received 8 November 2022; accepted after revision 8 February 2023; first published online 21 February 2023)

**Corresponding author** Chloé Yeung: Institute of Sports Medicine Copenhagen, Department of Orthopedic Surgery, Copenhagen University Hospital – Bispebjerg and Frederiksberg, Copenhagen, Denmark. Email: chloe.yeung@gmail.com

**Abstract figure legend** The primary goal of this study was to establish whether human tendon tissue is a circadian clock. Limited by obtaining only two patellar biopsies from each subject, we designed a study taking the biopsies 12 h apart, once in the day and once in the night, and analysed the gene expression by RNA sequencing. We found 280 time-dependent RNAs between 09.00 and 21.00 h biopsies and found night reduction in genes involved in collagen I fibrillogenesis by RT-qPCR. We also took samples to examine the presence of a chronomatrix by transmission electron microscopy but the differences were not statistically significant. In chronic patellar tendinopathy biopsies we found fewer time-dependent genes and no night reduction in expression of genes involved in collagen I fibrillogenesis. Our data suggest that tendinopathy may be accompanied by a dampened circadian rhythm.

**Key points**

- Tendinopathy is a major clinical problem with unsolved pathogenesis.
- Previous work in mice has shown that a robust circadian rhythm is required for collagen homeostasis in tendons.
- The use of circadian medicine in the diagnosis and treatment of tendinopathy has been stifled by the lack of studies on human tissue.
- Here, we establish that the expression of circadian clock genes in human tendons is time dependent, and now we have data to corroborate that circadian output is reduced in diseased tendon tissues.
- We consider our findings to be of significance in advancing the use of the tendon circadian clock as a therapeutic target or preclinical biomarker for tendinopathy.

## Introduction

Tendons are remarkable tissues connecting and transmitting force from muscle to bone, but they are also vulnerable to injury. Tendon overuse disorders (tendinopathy) in both upper and lower extremities affect ∼16.5 million people in the USA every year, and represent both a major clinical problem for the individual and a significant socio-economic cost (James et al., 2008). Although it is believed that tendinopathy is caused by inadequate homeostasis in response to overloading (Magnusson et al., 2010), the pathogenesis of tendinopathy has currently not been fully elucidated (Millar et al., 2021).

Many biological processes at the cell, tissue and organism levels are temporally coordinated in a 24-h fashion by the circadian clock. Circadian clock-controlled genes (CCGs) are highly tissue-specific, and represent up to 44% of the transcriptome in some tissues (visceral fat), with ∼12% of these encoding known drug targets (Dudek & Meng, 2014; Ruben et al., 2018). Chronic disruption to the circadian rhythm in humans (e.g. shift work) is a key factor in the development of some cancers and metabolic diseases (Kramer et al., 2022; Roenneberg & Merrow, 2016). Interestingly, tendons from mice with mutations in circadian clock genes exhibit phenotypes that resemble tendinopathy (Chang et al., 2020), supporting an important role for the circadian clock in maintaining a healthy tendon. Further, in humans with rheumatoid arthritis it has been shown that joint stiffness was reduced the most by the administration of systemic glucocorticoids during the night, which suppresses activation of the immune system that is pronounced during sleep (Paolino et al., 2017; Ursini et al., 2017). These data support the importance of circadian regulation of musculoskeletal tissues.

Extracellular matrix (ECM) homeostasis by resident fibroblasts is crucial to sustain a healthy tendon. We have recently demonstrated that the tendon circadian clock in mice regulates collagen secretion. The murine tendon ECM exhibits diurnal changes in ultrastructure [observed as fluctuations in the number of small-diameter (∼50 nm) collagen fibrils] and viscoelastic properties (higher energy dissipated during cyclic loading and quicker relaxation at night) and a higher abundance of non-covalently bound, processed collagen I during the day (Calverley et al., 2020; Chang et al., 2020). Together, these ECM changes represent the 'chronomatrix' in tendon. Further, there is evidence that human tendon is a peripheral clock tissue; isolated tenocytes in culture have demonstrated endogenous circadian rhythms (Yeung et al., 2014), circadian genes are expressed *in vivo* (Yeung et al., 2019) and there is reduced stiffness in the evening *in vivo* (Onambele-Pearson & Pearson, 2007; Pearson & Onambele, 2005, 2006). However, whether there is time-dependent circadian gene expression or the presence of a chronomatrix in human tendons *in vivo* and whether circadian gene expression is disrupted with chronic tendinopathy are currently not known. Such knowledge will be important in a clinical setting, as efforts to normalise potentially disrupted circadian gene expression, for example through pharmacological intervention, could provide new pathways to counteract and treat tendinopathy in humans.

We addressed these issues, and studied the gene expression, collagen fibril diameter distribution and proportion of acid-soluble collagen of human patellar tendon biopsies obtained at two time points, 12 h apart, from the same individuals. Furthermore, we investigated if there is a change in the expression of time-dependent genes in human tendons with chronic tendinopathy in different individuals. For RNA and acid-soluble collagen content analyses, patellar tendons were biopsied at 09.00 and 21.00 h to minimise interference to an individual's circadian rhythm, that is waking up in the middle of the night. For fibril diameter distribution analysis, patellar tendons were biopsied at zeitgeber time 10 (ZT10; 10 h into daylight) and then at ZT22 (16.00 and 04.00 h, respectively), which were based upon time points where the greatest difference was found in the percentage of narrow-diameter collagen fibrils in mouse tendons (Chang et al., 2020).

## Materials and Methods

### Experimental design

The primary objective was to establish whether there is time-dependent circadian gene expression in human patellar tendon. The secondary objectives were to determine the presence of a chronomatrix in human tendons and to investigate whether time-dependent gene expression changes in chronic tendinopathy. Participants with and without unilateral/bilateral chronic tendinopathy were recruited for patellar tendon biopsies at two time points, 12 h apart. Data were generated by RNA-sequencing (RNAseq) and RT-qPCR. RNA quality was assessed. Protein was precipitated from the material resulting from RNA isolation and the percentage of acid-soluble collagen was quantified by a hydroxyproline assay. Fibril diameter measurements were generated from transmission electron microscopy (TEM) images. For all experiments, replicate numbers are outlined below or in the figure legends. Minimum sample size was $n = 3$ biological samples, which is from different individuals per experimental groups. No power calculations were performed because this was an exploratory study. RNA analysis and collagen content experiments were blinded to biopsy times due to randomisation of the first biopsy. Collagen fibril diameter measurements were not blinded but semi-automated analysis was used to alleviate bias.

### Ethics

Written informed consent was obtained from all tissue donors (ethics approval H-42012152, amendment 43293 for the biopsies obtained for fibril size analysis, and ethics approval H-16019857, amendment 67 224 for the biopsies obtained for RNA and collagen content analyses) by the Regional Ethical Committee for the Hospital Region of Greater Copenhagen, in accordance with the *Declaration of Helsinki II*). The study was reported to the Danish register (Datatilsynet) and was performed in accordance with Danish law (Lov om behandling af personoplysninger).

### Human participants

Subjects were included via social media. For healthy patellar tendon biopsies, healthy male participants aged 18–40 years, who were not taking any medication, and did not have prior injuries to their lower extremities, including the knee joints, were recruited. A total of 17 participants were recruited for the analysis of RNA and collagen content and 10 participants were recruited for the analysis of collagen fibril diameter (Table S1). The 17 healthy participants for RNA were biopsied at 09.00 and 21.00 h and the 10 healthy participants for TEM were biopsied at 04.00 and 16.00 h. For tendinopathic tendon biopsies, otherwise healthy male and female participants (aged 20–55 years, body mass index between 18.5 and 30) with unilateral or bilateral chronic patellar tendinopathy for more than 3 months, who had not taken steroids or had had a steroid injection in the previous 6 months, and who did not have

an autoimmune disorder were recruited. A diagnosis of chronic tendinopathy was confirmed by a medical doctor via a physical exam. Chronic patellar tendinopathy was considered if participants reported activity-related pain, as well as pain upon palpation in the proximal part of the tendon, and was confirmed via ultrasound exam with at least one of the following indications: (i) thickening of the anterior–posterior diameter of at least 1 mm compared to the mid-tendon level, (ii) increased power doppler signal and (iii) a hypoechogenic area corresponding to the symptomatic area of the tendon. Eight men and two women were recruited for biopsies of both patellar tendons at either 09.00 or 21.00 h. All participants were asked how many hours per week they spend on physical activity and patients with chronic tendinopathy were additionally asked how many hours they used to spend on physical activity.

## Tendon biopsies

Biopsies were obtained as described previously (Yeung et al., 2019). In brief, the skin over the biopsy site was sterilised and treated with local anaesthetics (1% lidocaine). The biopsies were taken from the patellar tendon with a Bard Magnum Biopsy Instrument (C.R. Bard Inc., Covington, GA, USA) and a 14 G needle. For the RNA and collagen content analyses, healthy participants were randomised to have their first biopsy taken at 09.00 or 21.00 h, and whether it was from the dominant or non-dominant leg. For the TEM, the first leg (dominant or non-dominant) to be biopsied was alternated between participants at 16.00 h. The second biopsy was taken from the other leg at 04.00 h. The 04.00 h biopsies were taken as second biopsies to avoid the possibility of a disrupted sleep or circadian rhythm impacting on the next biopsy. Patients with chronic tendinopathy were randomised to have their biopsies taken at 09.00 or 21.00 h, first from the right leg and then from the left leg. The weights of the biopsies were ∼5–10 mg, cleared of adipose tissue and blood, and snap frozen immediately with liquid nitrogen and stored at −80°C until RNA isolation or placed immediately into 2% glutaraldehyde in 100 mM phosphate buffer for preparation for TEM analysis. Unfortunately, it is not feasible to perform longitudinal time-series studies on human tendon. One of the reasons is that a single tendon biopsy causes local tissue trauma that persists for several months (Heinemeier et al., 2016), and therefore only two biopsies from the same tendon type can be obtained from any single individual.

## Tendon tissue culture

Primary human tendon fibroblasts were prepared from healthy gracilis and semitendinosus tendon (waste tissue obtained in connection with anterior cruciate ligament reconstruction surgery) as previously described (Yeung et al., 2020). Tendon fibroblasts were cultured in DMEM/F-12 medium supplemented with 10% fetal calf serum (FCS) and 50 U/ml penicillin and 50 $\mu$g/ml streptomycin (complete medium) at 37°C in 5% $CO_2$. For the time-series RNA isolation, $1 \times 10^4$ cells/cm$^2$ were plated onto 35 mm dishes and cultured in complete medium for 2 days. Then, cells were synchronised using 100 nM dexamethasone (Sigma-Aldrich, St Louis, MO, USA) in complete medium for 24 h, after which the medium was exchanged for complete medium. RNA was isolated every 4 h from 12 to 56 h after the medium change.

## Zeitgeber time conversion

ZT was calculated for each sample from the time of sunrise on the calendar date of the biopsy. The 17 biopsies for RNA and collagen content analyses were collected at different times during the year: February ($n = 4$), end of May to beginning of June ($n = 4$), August ($n = 1$), November ($n = 1$) and December ($n = 7$) in Copenhagen, Denmark. Based on ZT times, the biopsies were divided into three groups: Group 1: ZT1 *vs.* ZT13 ($n = 7$; range: 0.45–1.00 h and 12.18–12.85 h, respectively), Group 2: ZT2 *vs.* ZT14 ($n = 5$; range: 1.23–1.87 and 12.97–13.93 h, respectively) and Group 3: ZT4 *vs.* ZT16 ($n = 5$; range: 3.75–4.63 and 15.57–16.47 h, respectively). The one sample with an evening biopsy ZT of 12.97 h was put in Group 2 and not Group 1 because the ZT of the morning biopsy (1.23 h) was closer to the morning biopsy ZT range of Group 2. The biopsies for TEM analysis were collected during 10 days in August in Copenhagen, Denmark, and 16.00 and 04.00 h were determined as the wall clock time points for biopsy collection from calculating when ZT10 and ZT22 were on those days. The tendinopathic and contralateral biopsies were collected in November ($n = 4$), December ($n = 5$) and February ($n = 1$) in Copenhagen, Denmark. Sunrise times were obtained from the webpage https://www.sunrise-and-sunset.com/en/sun/denmark/copenhagen.

## RNA isolation, quantification and quality assessment

RNA was isolated from tendon cells and tendon biopsies as described in detail previously (Heinemeier et al., 2013). RNA concentration was measured using a Quant-iT RiboGreen RNA Assay Kit (Thermo Fisher Scientific, Waltham, MA, USA) following the manufacturer's instructions. RNA quality was assessed using an RNA 6000 Pico Kit (Agilent) and Eukaryote Total RNA Pico Assay on a Bioanalyzer (Agilent, Santa Clara, CA, USA).

## RNA sequencing

Due to low RNA concentration and sample degradation, only five pairs of healthy tendon samples were analysed by RNAseq. Due to difficulty in recruiting participants, the samples were sequenced in two batches (Batch 1 $n = 3$, Batch 2 $n = 2$). Due to error with cDNA library preparation, one 21.00 h chronic tendinopathy sample was not analysed by RNAseq. cDNA library preparation and RNAseq were performed at Genewiz GmbH (Leipzig, Germany). RNAseq library preparation was done using NEBNext Ultra II Directional RNA Library Prep Kit for Illumina following the manufacturer's instructions (New England Biolabs, Ipswich, MA, USA). Total RNA was first enriched for mRNA by either rRNA depletion (Batch 1) using NEBNext rRNA Depletion Kit (New England Biolabs) or mRNA enrichment with Oligo(dT) beads (Batch 2 and tendinopathy samples). Briefly, enriched mRNAs were fragmented. First- and second-strand cDNA were subsequently synthesised. The second strand of cDNA was marked by incorporating dUTP during the synthesis. cDNA fragments were adenylated at 3′ ends, and indexed adapter was ligated to cDNA fragments. Limited cycle PCR was used for library amplification. The dUTP incorporated into the cDNA of the second strand enabled its specific degradation to maintain strand specificity. Sequencing libraries were validated using DNA Kit on the Agilent 5600 Fragment Analyzer (Agilent Technologies, Palo Alto, CA, USA), and quantified by using a Qubit 4.0 Fluorometer (Invitrogen, Carlsbad, CA, USA). The sequencing libraries were multiplexed and clustered on the flowcell. After clustering, the flowcell was loaded on the Illumina NovaSeq 6000 instrument according to the manufacturer's instructions. The samples were sequenced using a $2 \times 150$ pair-end (PE) configuration (∼50 million reads per sample). Image analysis and base calling were conducted by the NovaSeq Control Software v1.6 on the NovaSeq instrument. Raw sequence data (.bcl files) generated from Illumina NovaSeq was converted into fastq files and de-multiplexed using Illumina bcl2fastq program version 2.20. One mismatch was allowed for index sequence identification.

## RNAseq statistical analysis

Reads were aligned to the Human GRCh38 (release 34) assembly and transcripts (exons) were counted using SubRead v1.6.2 (http://www.ncbi.nlm.nih.gov/pubmed/23558742). In total, 19−48 million counts were obtained in each sample with 31 000−37 000 different transcripts detected. Count normalisation and differential analyses were performed using DESeq2 v1.34.0 (Love et al., 2014). For the paired healthy samples, subject ID was included in the model to account for differences between subjects and cDNA library preparation, but otherwise only the specific test type was included in each model. To account for log-fold inflation of the low counts, the lcfshrink method of DESeq2 was used. The resultant shrunken lc2fc, *p*-values and adjusted *P*-values [false discovery rate (FDR)] are provided in Data file S1 for healthy tendons and Data file S5 for chronic tendinopathy together with the normalized and raw counts. Functional enrichment analysis was performed using the online tool DAVID version 6.7 (Huang da et al., 2009).

## Real-time RT-qPCR

Due to insufficient leftover RNA, samples used for Batch 1 RNAseq were not analysed by RT-qPCR. The amount of mRNA was measured by RT-qPCR and normalised to the amount of 60S acidic ribosomal protein P0 (*RPLP0*) mRNA, a reference gene that is stable for circadian studies (Kosir et al., 2010). Primer design and RT-qPCR was performed as previously detailed (Doessing et al., 2010; Yeung et al., 2019), with the exception that cDNA synthesis was performed with 30 ng total RNA using qScript (QuantaBio, Beverly, MA, USA) as recommended in the kit, including the oligo-dT and random primer blend. Standard curves with known number of molecules of oligonucleotides identical to the expected product was used for each PCR. Primer sequences used were:

*BMAL1* (NM_001030272.2; GGGCTGGGGCAGGAAAA ATAG, GAGCCACAGCTAGAAGGCGATG),
*CLOCK* (NM_001267843.1; GCCCAACCCCTTCTGCCT CTTC, CGTCGGGATCTTGGTTGGTGT),
*COL1A1* (NM_000088.3; GGCAACAGCCGCTTCACCT AC, GCGGGAGGTCTTGGTGGTTTT),
*COL1A2* (NM_000089.3; AGGGCAACAGCAGGTTCAC TTACA, GGGCAGGCGTGATGGCTTATTT),
*COL4A1* (NM_001846.3; TCGCTGTGGATCGGCTACTC TT, CGATGAATGGCGCACTTCTAAAC),
*COL5A1* (NM_000093.4; AGCAGATGAAACGGCCCC TG, TCCTTGGTTAGGATCGACCCAGT),
*CRY1* (NM_004075.4; GCAGATGTGTTTCCCAGGCTT TTC, TAGCTGCGTCTCGTTCCTTTCC),
*GAPDH* (NM_002046.6; CCTCCTGCACCACCAACTG CTT, GAGGGGCCATCCACAGTCTTCT),
*NR1D1* (NM_021724.4; GCAAGAGCACCAGCAACAT CAC, GCAACGTCCCCACACACTTTACAC),
*MMP14* (NM_004995.4; CCTACCGACAAGATTGATGC TGCT, TCCACTGCCCTGAGCTCTTCGT),
*PER2* (NM_022817.2; AACCAGCCCACCTGCTCCTACC, GCTGGGAACTCGCATTTCCTCTT),
*RPLP0* (NM_053275.3; GGAAACTCTGCATTCTCGCT TCCT, CCAGGACTCGTTTGTACCCGTTG).

## Precipitation of protein and solubilisation with acetic acid

Protein was precipitated from the interphase and organic phase material resulting from RNA isolation according to the TRI Reagent protocol. In brief, proteins were mixed with 100% ethanol and centrifuged for 5 min at 2000 $g$, 4°C. Proteins from the supernatant were then precipitated with 2-propanol and centrifuged for 10 min at 12 000 $g$, 4°C. Protein pellets were washed with 0.3 M guanidine hydrochloride in 95% ethanol for 20 min at room temperature (RT) and then centrifuged at 7500 $g$, 4°C. The wash step was performed six times. Washed protein pellets were stored in 100% ethanol at −80°C. Protein pellets were transferred to hydrolysis tubes and dried under a nitrogen flux and then incubated with 500 $\mu$l ice-cold 0.5 M acetic acid on a rotation device overnight at 4°C. The samples were then centrifuged for 1 h at 12 000 $g$, at 4°C. Then, 250 $\mu$l of the supernatant (50% of acid-soluble proteins) was carefully transferred to hydrolysis tubes. An equal volume of 12 M HCl was then added to both the supernatant and the tube containing the remaining supernatant and pellet (50% of acid-soluble proteins and 100% of non-acid-soluble proteins), vortexed to mix and incubated overnight at 110°C. The tubes were then cooled and briefly centrifuged. Hydrolysed samples were dried on a heat block at 37°C under a nitrogen flux. The collagen content in the dried samples was determined by a hydroxyproline assay.

## Hydroxyproline assay

A hydroxyproline assay was performed as described previously (Svensson et al., 2018). Two technical replicates were made for all standards and samples. In brief, dried hydrolysed acid-soluble and non-acid-soluble protein fractions were thoroughly reconstituted with 310 $\mu$l acetate–citrate buffer (0.55 M acetate, 0.13 M citrate, pH 6.0). Dilutions (1:5 and 1:20) were made for the tubes that contained the remaining supernatant and pellet. Tubes that contained only the supernatant were not diluted. In duplicates, 75 $\mu$l oxidizing reagent (50 mM chloramine T in 50% 1-propanol) was added to 150 $\mu$l standards/samples, vortexed to mix and incubated for 20 min at RT. Then, 75 $\mu$l colour reagent (1.0 M 4-dimethylaminobenzaldehyde, 15.4% perchloric acid in 70% 1-propanol) was added to each sample, vortexed and incubated for 25 min in a water bath at 60°C. Reactions were then stopped by incubating samples on ice for 2 min. After this, 200 $\mu$l of each sample was transferred to a transparent assay plate and the absorbance at 570 nm was measured. The hydroxyproline content of each sample was calculated against a known standard curve. Collagen content was calculated assuming 13.5% hydroxyproline by mass (Neuman & Logan, 1950). The amount of acid-soluble collagen was determined by multiplying the collagen content originally in the supernatant (50% of acid-soluble proteins) by 2 and expressed as a percentage of the sum of collagen content originally in the supernatant (50% of acid-soluble proteins) and in the remaining supernatant and pellet (50% of acid-soluble proteins and 100% of non-acid-soluble proteins).

## Transmission electron microscopy

For TEM, tissues were prepared as described previously (Starborg et al., 2013). In brief, tissues were fixed in 1% osmium and 1.5% potassium ferrocyanide in 0.2 M sodium cacodylate buffer for 1 h, washed with distilled water, then incubated with 1% tannic acid in 0.1 M cacodylate buffer for 1 h, washed with distilled water, and then incubated with 1% osmium in water for 30 min. Samples were then washed with distilled water and stained with 1% uranyl acetate in water for 1 h, then dehydrated in acetone and embedded in resin. Ultrathin sections were mounted onto copper grids. Grids were examined with a Tecnai 12 instrument (FEI, Hillsboro, OR, USA) fitted with a 2k × 2k-cooled Teitz F214A CCD camera (Tietz Video and Image Processing Systems).

## Semi-automated fibril diameter measurements

Due to technical issues, one day sample and two night samples were not analysed. Fibril diameters were measured with the aid of a custom MATLAB script (MATLAB 2017b, The MathWorks Inc., Natick, MA, USA) that automatically detected and measured the diameter of fibril cross-sections using a circular Hough transform (Smereka & Dulęba, 2008). In each image, a region of collagen fibrils was selected, free of cracks, cells or other larger non-collagenous structures. An unbiased counting frame with guard regions was superimposed on the selected region and fibrils were automatically detected and measured. The results were subsequently curated manually to remove or add any incorrect or missing fibril measurements. Finally, fibrils touching the exclusion borders of the counting frame were removed to obtain the correct counting statistics. Raw data outputs can be found in Data file S4.

## Statistical analysis

Mean, SD and SEM were calculated for datasets from at least three biological replicates ($n$ numbers are indicated in figure legends). SEM bars are plotted. Time-series RT-qPCR for synchronised cells ($n$ = 3 biological replicates) were plotted separately for each cell preparation. Statistical tests used, significant differences and $P$ values are indicated in the figures or noted in

the corresponding figure legends and reported in the Statistical Summary Document. MetaCycle (Wu et al., 2016) was applied to log-transformed RT-qPCR data from synchronised tendon fibroblasts to determine circadian rhythmicity.

## Results

### Time-dependent expression of conserved circadian genes in human tendon

Seventeen healthy individuals participated (Fig. 1A). Delays in recruitment meant that biopsies were collected during different seasons throughout the year (Table S1). To account for seasonal variance ZT (hours after sunrise) was calculated for each individual before data analysis and individuals were grouped accordingly

(Table S1). To establish if circadian clock gene expression was time-dependent in human tendons, we performed RNAseq on human patellar tendon tissue biopsied from five individuals at two time points, 12 h apart (09.00 and 21.00 h), as close to the wall clock time as possible (Table S1). A total of 43 792 different RNAs were detected of which 33 292 were detected in at least half the samples and used for differential expression analyses. Of those, 17 152 were protein-coding RNAs. There were 280 RNAs that were differentially expressed between the two time points [$P$(FDR) < 0.05; Data file S2]. The majority of these RNAs were protein-coding (120 upregulated at 09.00 h, 86 upregulated at 21.00 h) and ∼20% were long non-coding RNAs (Fig. 2A). Functional enrichment analysis of all significant RNAs revealed only two functional terms with enrichment scores ≥2 and these were circadian rhythm and transcription (Data file S3). Note that only one gene

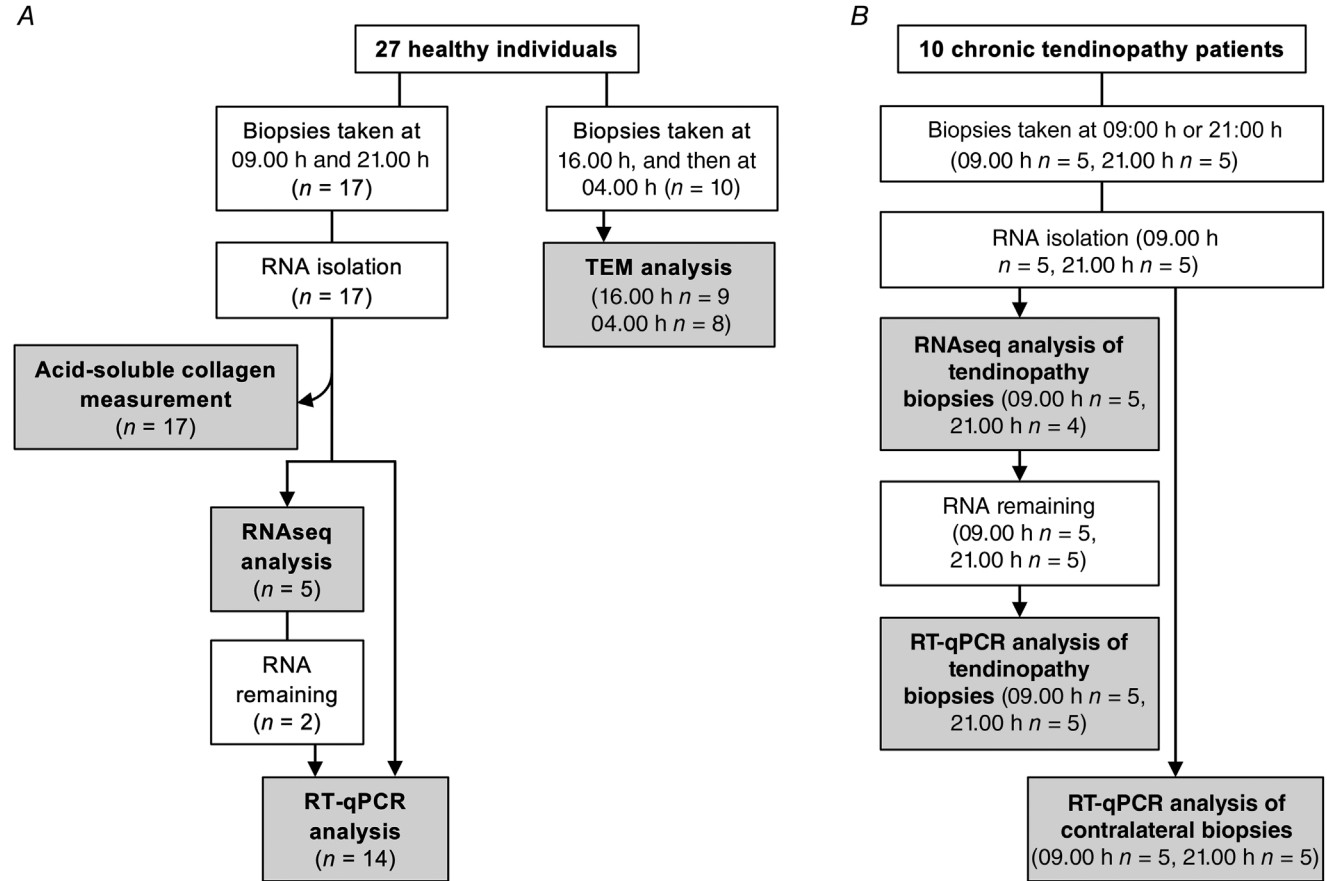

**Figure 1. Sumary of tendon biopsies used in the study**
*A*, a total of 27 healthy individuals were recruited. Patellar biopsies taken from 17 individuals at 09.00 h and 21.00 h were used for RNA analyses and acid-soluble collagen measurement, and biopsies taken from 10 individuals at 16.00 h and then at 04.00 h were used for TEM analysis. *B*, ten patients with chronic patellar tendinopathy were recruited. Patellar biopsies from the tendinopathic and contralateral tendons were taken from five individuals at 09.00 h and from another five individuals at 21.00 h. Tendinopathic biopsies were used for RNAseq and RT-qPCR analyses. Contralateral biopsies were used for RT-qPCR analysis. Details of biological replicates used can be found in Table S1.

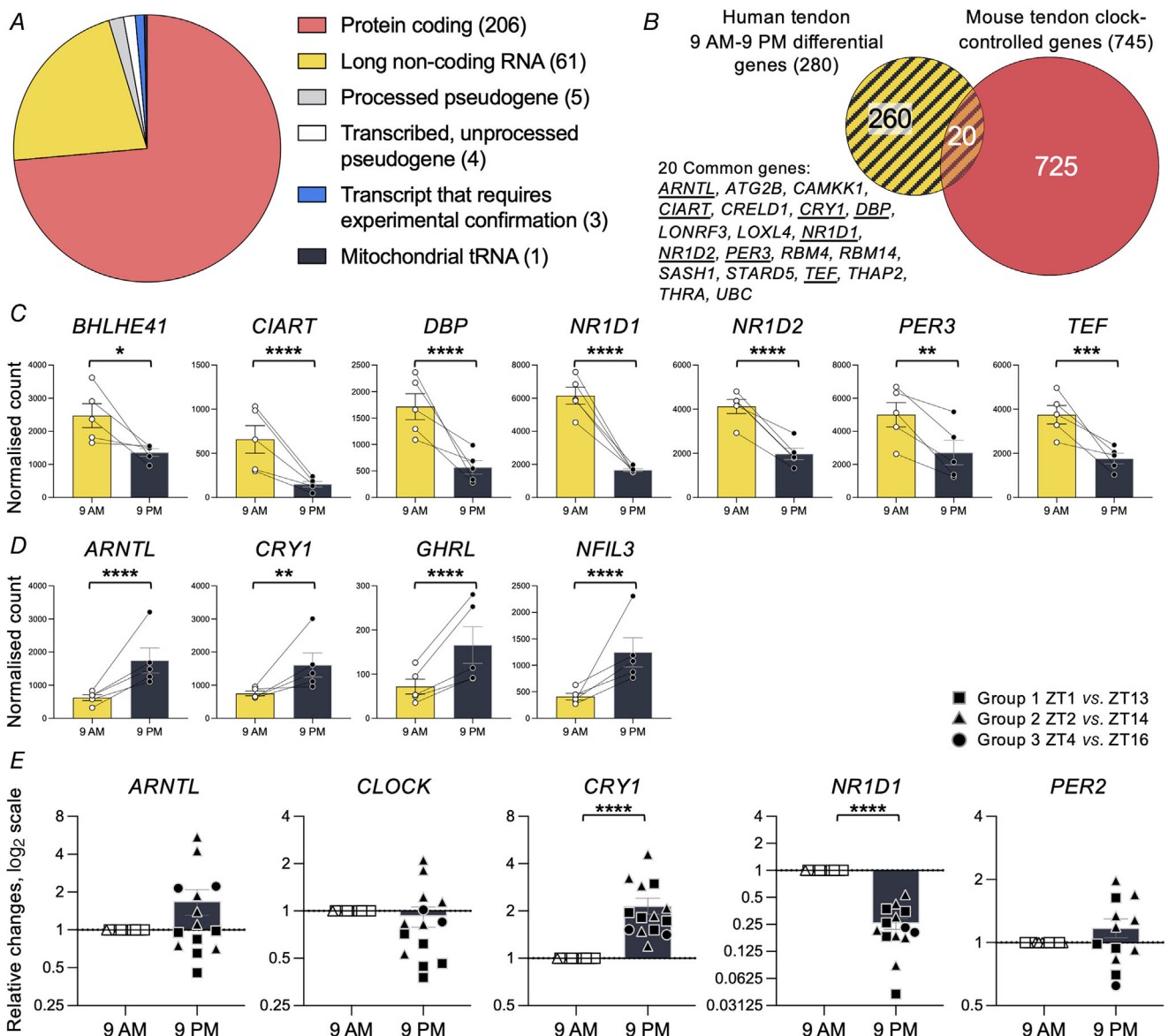

**Figure 2. Time-dependent expression of conserved circadian genes in human patellar tendon**
Patellar tendon biopsies taken at 09.00 and 21.00 h were analysed by RNAseq and the expression of 280 RNAs was statistically different between day and night [*P* (FDR) < 0.05; *n* = 5]. *A*, summary of the types of RNA that were differentially expressed between 09.00 and 21.00 h. Numbers in parentheses indicate the number of RNAs in each group. *B*, comparison of the protein-coding genes in human tendon with reported mouse tendon circadian transcriptome (Yeung et al., 2014) with 20 common genes. Circadian clock genes are underlined. *C*, normalised RNA counts for conserved circadian clock genes or direct clock output genes significantly upregulated in the 09.00 h biopsies – *BHLHE41*, *CIART*, *DBP*, *NR1D1*, *NR1D2*, *PER3* and *TEF* [*n* = 5; bars show SEM; connecting lines indicate paired samples; *P* (FDR) = 0.0435, 4.2E-23, 7.3E-5, 2.4E-23, 5.7E-8, 0.0015, 8.4E-4, respectively, from DESeq2 analysis]. *D*, normalised RNA counts for conserved circadian clock genes or direct clock output genes significantly upregulated in the 21.00 h biopsies – *ARNTL*, *CRY1*, *GHRL* and *NFIL3* [*n* = 5; bars show SEM; connecting lines indicate paired samples; *P* (FDR) = 7.5E-9, 0.0030, 5.4E-5, 2.5E-5, respectively, from DESeq2 analysis]. Connecting lines indicate paired samples. *E*, change to expression of circadian clock genes in all the healthy biopsies relative to 09.00 h analysed by RT-qPCR. Expression was normalised to *RPLP0*, and expressed in $\log_2$ values, relative to the day biopsy value (*n* = 14 except *n* = 12 for PER2; bars show SEM; ****P* < 0.0001, from paired *t* tests performed on the log-transformed data). [Colour figure can be viewed at wileyonlinelibrary.com]

showed differential expression due to biopsy order: sterol *O*-acyltransferase 2 (*SOAT2*) [*P* (FDR) < 0.05; Data file S2].

We compared the list of time-dependent RNAs to circadian clock-controlled genes reported for mouse tendon (Yeung et al., 2014) and found 20 overlapping genes (Fig. 2*B*). Eight of these overlapping genes encode conserved circadian clock proteins or direct clock outputs (Anafi et al., 2017; Wu et al., 2018) (Fig. 2*B*). In total we found 11 conserved circadian genes. The expression of *BHLHE41* (basic helix–loop–helix family member E41), *CIART* (circadian-associated repressor of transcription), *DBP* (D-box binding PAR BZIP transcription factor), *NR1D1* (nuclear receptor subfamily 1 group D member 1), *NR1D2* (nuclear receptor subfamily 1 group D member 2), *PER3* (period 3) and *TEF* (thyrotroph embryonic factor) was upregulated in the day (Fig. 2*C*). Additionally, the expression of *ARNTL* [aryl hydrocarbon receptor nuclear translocator like-1, also known as brain and muscle ARNT-like 1 (BMAL1)], *CRY1* (cryptochrome circadian regulator 1), *GHRL* (ghrelin and obestatin prepropeptide) and *NFIL3* (nuclear factor, interleukin 3 regulated) were upregulated at night (Fig. 2*D*). We validated the expression of circadian genes that were identified as significant and two other non-changing genes [clock circadian regulator (*CLOCK*) and *PER2*] by RT-qPCR in additional biopsies and confirmed a significant difference for the expression of *CRY1* and *NR1D1* ($n = 14$ except $n = 12$ for *PER2*; Fig. 2*E*). Biopsies were taken at 09.00 or 21.00 h, but as time of sunrise varies over the year, we also analysed the samples according to similar ZT (Table S1). In addition to *CRY1* and *NR1D1*, the expression of *CLOCK* was found to be significantly downregulated from ZT2 to ZT14 ($n = 5$; Fig. S1). Together, these data suggest that obtaining biopsies from two time points was sufficient for establishing time-dependent expression of circadian clock genes in human patellar tendon.

### Day-to-night collagen I expression differences *in vivo*

When genes related to collagen I fibrillogenesis were analysed as a function of time by RT-qPCR, collagen α1(I) (*COL1A1*) and collagen α2(I) (*COL1A2*) were significantly reduced from 09.00 to 21.00 h ($n = 14$; Fig. 3*A*). There was also a tendency for reduced night expression of collagen α1(V) (*COL5A1*), which is required for collagen fibril nucleation (Wenstrup et al., 2004) (Fig. 3*A*). Analysis of the ZT2 *versus* ZT14 biopsies confirmed significant differences for *COL1A2* and *COL5A1*, and a strong trend for time-dependent differences was observed for *COL1A1* and *MMP14*, the latter of which is required for the release of newly assembled collagen fibrils from the cell (Taylor et al., 2015) ($n = 5$; Fig. S1).

A significant circadian rhythm was not observed for collagen I transcripts in a previous mouse tendon screen (Chang et al., 2020). Therefore, to establish whether *COL1A1*, *COL1A2* and *COL5A1* transcripts were regulated by the endogenous tendon circadian clock in human tissues, primary tendon fibroblast cultures were synchronised and time-series RT-qPCR was performed. The rhythmic expression of *CRY1* and *NR1D1* with anti-phase expression patterns to one another confirmed that the synchronised tendon cells retained their endogenous circadian rhythms, but no rhythmicity in the expression of *GAPDH*, *COL1A1*, *COL1A2* or *COL5A1* was observed ($n = 3$ except $n = 2$ for the 12 h time point; Fig. 3*B*). These data suggest that, like mouse tendons, the expression of collagen I genes in human tendons is not under direct circadian control, but expression levels may be influenced by additional day–night factors *in vivo*.

### Day-to-night collagen ECM changes are not apparent in human tendons

To establish the presence of a chronomatrix in human tendons, we used the protein precipitated after RNA isolation from the same day and night biopsies. We performed acid solubilisation to extract the pool of newly synthesised, non-covalently bound and immature cross-linked collagens, and quantified it using a hydroxyproline assay. We detected $2.1 \pm 0.8\%$ (SEM) acid-soluble collagen in the 09.00 h biopsies and a mean of $1.5 \pm 0.5\%$ (SEM) acid-soluble collagen in the 21.00 h biopsies ($n = 17$; Fig. 4*A*). Analysing the biopsies in groups according to ZTs also showed no significant difference between the day and night acid-soluble collagen fractions ($n = 7$ for Group 1, $n = 5$ for Groups 2 and 3; Fig. S2). Additionally, there was no correlation between ZT and the percentage of acid-soluble collagen (data not shown).

The chronomatrix in mouse tendon is also characterised by a diurnal change in narrow (∼50 nm) diameter fibrils (Chang et al., 2020). We performed TEM and collagen fibril diameter measurements on human patellar tendon tissue biopsied from 10 individuals at 16.00 and 04.00 h (which corresponded to ZT10 and ZT22, respectively, at the time of sampling), as close to the wall clock time as possible (Fig. 1*A*; Table S1). There appeared to be a trend of increased number of narrow (60–90 nm) diameter fibrils at 04.00 h (∼51%) than at 16.00 h (∼67%), but there was no statistical difference in the median fibril diameters or cumulative percentage of number of fibrils according to increased size between day and night biopsies ($n = 9$ for 16.00 h and $n = 8$ for 04.00 h; Fig. 4*B–D*; Fig. S3, Data S4).

## Disruption of time-dependent gene expression of circadian clock and collagen I genes in chronic tendinopathy

To determine whether the circadian rhythm in tendon is dysregulated in tendinopathy, RNAseq was performed on chronic tendinopathic tendon biopsies. Ten patients

with chronic tendinopathy participated (Fig. 1). Five patients had biopsies taken at 09.00 h and five patients were biopsied at 21.00 h, as close to the wall time as possible (Fig. 1*B*; Table S1). However, due to a problem with the cDNA library preparation, one chronic 21.00 h sample that was sequenced was excluded from the post-sequencing analyses. There was no significant

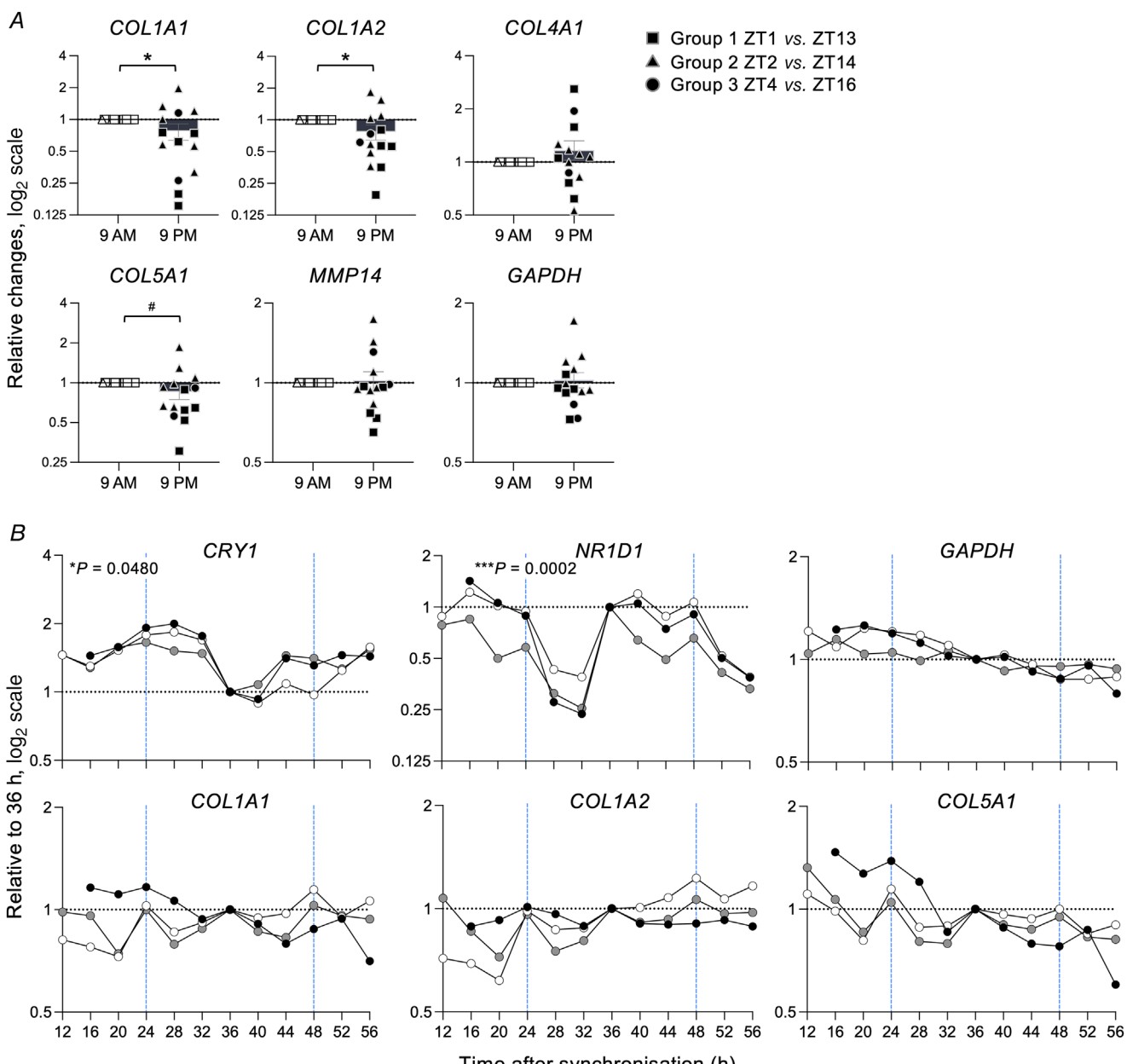

**Figure 3. Collagen I expression in human tendon is sensitive to time of day *in vivo***
*A*, change in gene expression in all the healthy biopsies relative to 09.00 h analysed by RT-qPCR. Expression was normalised to *RPLP0* and expressed in log₂ values [*n* = 14 biological samples; bars show SEM; **P* = 0.0308 (*COL1A1*), 0.0199 (*COL1A2*) and #*P* = 0.0542, from paired *t* tests performed on the log-transformed data]. *B*, gene expression in synchronised primary human tendon fibroblasts analysed by RT-qPCR. Expression was normalised to *RPLP0*, and expressed in log₂ values, relative to the 36 h time point value [*n* = 3 biological replicates except *n* = 2 for 12 h; **P* = 0.0480 (*CRY1*), ****P* = 0.0002 (*NR1D1*), from MetaCycle analysis performed on log-transformed data; dotted blue lines mark the beginning of a new 24 h period]. [Colour figure can be viewed at wileyonlinelibrary.com]

age difference between the participants biopsied on their tendinopathic patellar tendon at 09.00 h and those biopsied at 21.00 h ($n = 5$ for 09.00 h, $n = 4$ for 21.00 h; $P = 0.3175$, Mann–Whitney test; Table S1). A total of 44 205 different RNAs were detected, of which 32 319 were detected in at least half the samples and used for differential expression analyses. Of those, 16 967 protein-coding RNAs were detected, 10 008 of which were also detected in healthy biopsies. Only 23 RNAs were found to have time-dependent expression levels and 19 of these genes were protein-coding (Fig. 5*A*; Data file S5). Comparison of these time-dependent RNAs with those identified in healthy tendons revealed that only four RNAs maintained time-dependent expression (Fig. 5*B*). The expression of only three of the 11 circadian clock genes (*CIART*, *DBP* and *NR1D1*) that were previously identified in healthy tendon biopsies remained time-dependent with

chronic tendinopathy [$P$ (FDR) $< 0.05$; Data file S4]. Of note, there was no significant difference between the age of healthy controls and chronic tendinopathy patients ($n = 5$ for healthy, $n = 9$ for chronic tendinopathy; $P = 0.3666$, Mann–Whitney test). Together, these data suggest that the circadian rhythm may be disrupted, and clock outputs are reduced in chronic tendinopathy.

Contralateral biopsies were also taken from the patellar tendon of the non-chronic tendinopathy leg in the same subjects at the same time (Fig. 1; Table S1). Of the 10 contralateral biopsies, two had increased ultrasonographic Doppler flow (as a sign of hyper-vascularisation), two exhibited distal pain and two had diffused pain and the remaining four biopsies were from clinically asymptomatic tendons. RNA was isolated from these contralateral biopsies and analysed by RT-qPCR. There was a significant time difference in the expression

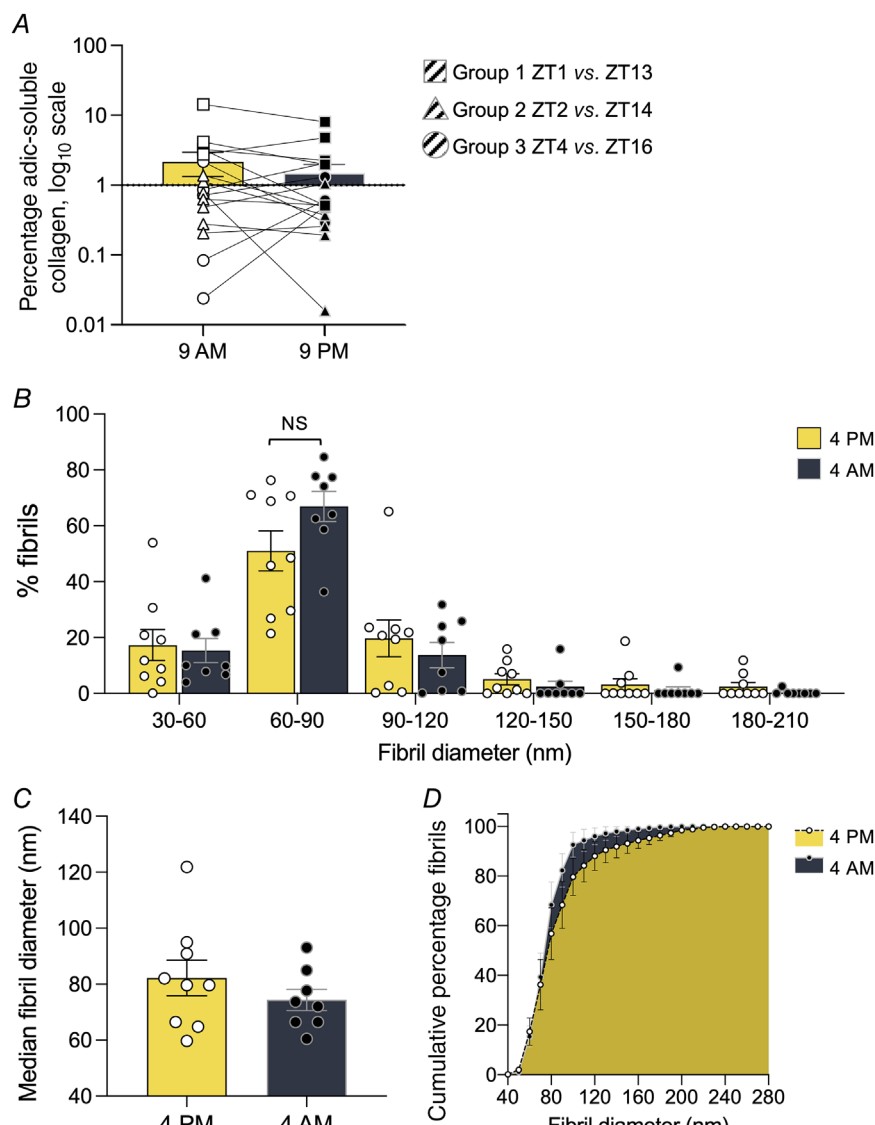

**Figure 4. Comparison of the acid-soluble collagen fraction and fibril diameter distribution in human tendon biopsies from day and night**
The percentage of acid-soluble collagen was determined in patellar tendon biopsies taken at 09.00 and 21.00 h. *A*, statistical analysis showed no significant difference between the amount of acid-soluble collagen in tendon tissues taken in the morning and the night ($n = 17$ biological samples; bars show mean; $P = 0.5392$, from paired *t* tests on log-transformed data). *B*, fibril diameter distribution was analysed in patellar tendon biopsies taken at 16.00 and 04.00 h. No significant differences were found between the percentage of fibrils in the respective 30 nm bins [$n = 9$ (16.00 h), $n = 8$ (04.00 h); bars show mean, minimum and maximum values; $P = 0.0965$ (60–90 nm), from unpaired *t* tests with Welch's correction; bars show mean ± SEM]. *C*, median diameter of fibrils was not significantly different between day and night biopsies [$n = 9$ (16.00 h), $n = 8$ (04.00 h); bars show mean, minimum and maximum values; $P = 0.3059$, from unpaired *t* test with Welch's correction]. *D*, cumulative percentage of fibril diameters measured was not significantly different between day and night biopsies [$n = 9$ (16.00 h), $n = 8$ (04.00 h); bars show SEM; $P = 0.2225$, from Kolmogorov–Smirnov test for positive changes from 16.00 h to 04.00 h; values are mean ± SEM. [Colour figure can be viewed at wileyonlinelibrary.com]

levels of *CRY1* and *NR1D1* but no significant differences between diseased and the contralateral tendons ($n = 5$; Fig. 5C). No significant time or health differences were found for the expression of *ARNTL*, *CLOCK*, *GAPDH*, *COL1A1*, *COL1A2* and *COL5A1* (Fig. 5C). To address whether a reduced activity level in chronic tendinopathy patients compared to healthy participants explained why no time-dependent significant differences were found in the expression of collagen genes, we compared the amount of time they spent on physical activity and found no significant difference [$n = 17$ for healthy and $n = 10$ for chronic tendinopathy; Fig. S4; $7.6 \pm 3.9$ h (SD) per week in healthy, $6.1 \pm 4.2$ h (SD) per week in tendinopathy]. However, we found that participants with tendinopathy

had significantly reduced their habitual physical activity when they began to develop symptoms [$n = 10$; Fig. S4; $9.5 \pm 7.8$ h/week (SD) before tendinopathy].

## No effect of leg dominance on clock gene expression or acid-soluble collagen content

Gene expression and collagen turnover in tendon responds to mechanical loading, so we wanted to rule out any effects due to differential loading on the two different legs. Therefore, we analysed the RNAseq and acid-soluble collagen data as a function of leg dominance in the healthy biopsies. We did not find any significant difference in the expression of clock genes ($n = 5$; Data file S2). We did,

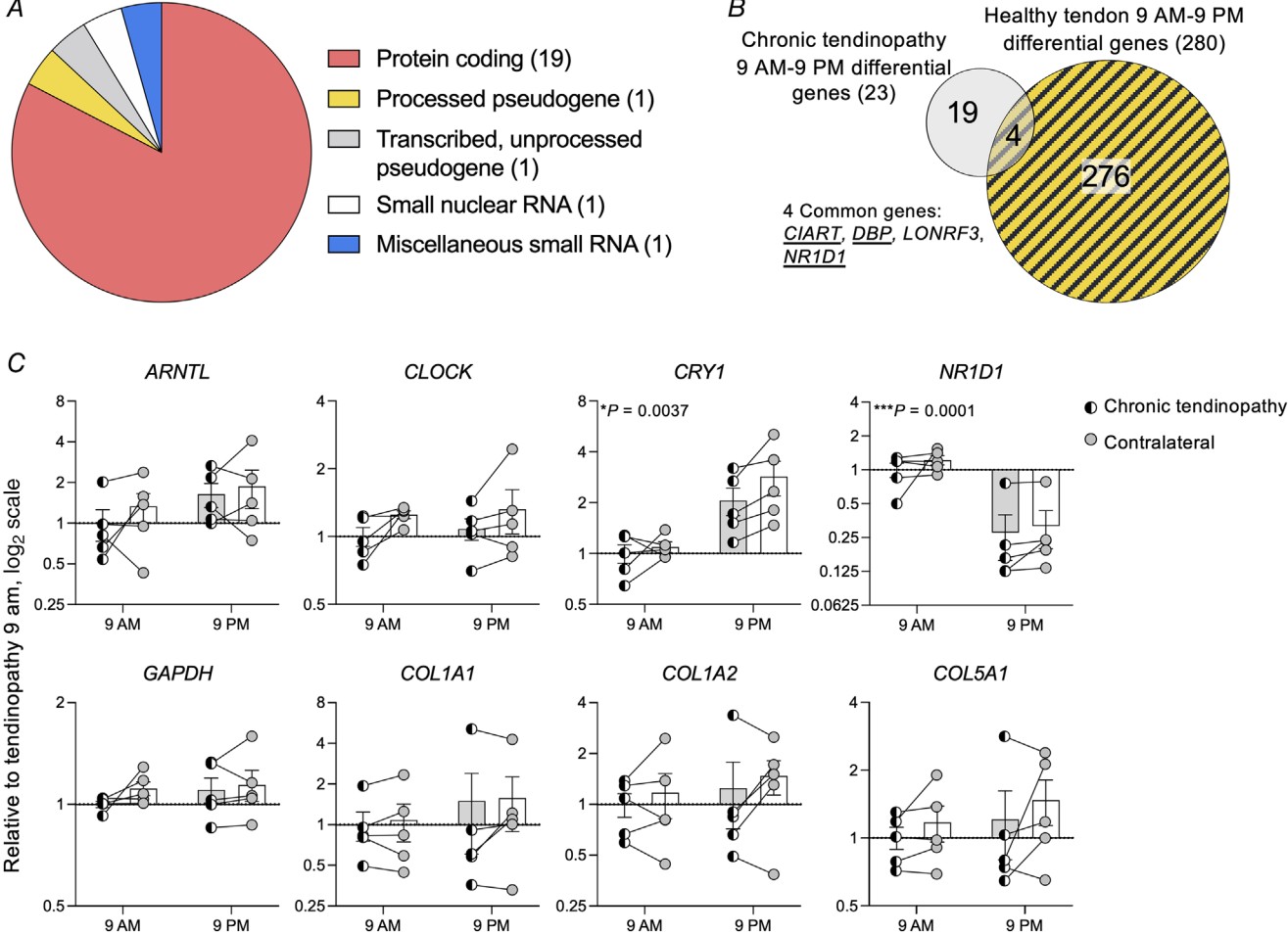

**Figure 5. Tendinopathic tendons lose time-dependent expression of many circadian clock genes**
Tendinopathic patellar tendon biopsies taken at 09.00 and 21.00 h were analysed by RNAseq and the expression of 23 RNAs was statistically different between day and night [$n = 4$; $P$ (FDR) $< 0.05$, from DESeq2 analysis]. *A*, summary of the types of RNA that were differentially expressed between 09.00 and 21.00 h. Numbers in parentheses indicate the number of RNAs in each group. *B*, comparison of time-dependent genes found in healthy and tendinopathic tendons showed five common genes. Circadian clock genes are underlined. *C*, changes in gene expression relative to the day tendinopathy biopsy value in chronic tendinopathy and contralateral tendons analysed by RT-qPCR. Expression was normalised to *RPLP0*, and expressed in log$_2$ values, relative to the day tendinopathy biopsy value [$n = 5$; bars show mean $\pm$ SEM; *$P = 0.0037$ (*CRY1*) and ***$P = 0.0001$ (*NR1D1*) for time effect, from two-way ANOVA performed on the log-transformed data]. [Colour figure can be viewed at wileyonlinelibrary.com]

however, identify nine genes that were differentially expressed between the dominant and non-dominant leg, including *COL1A2* [$P$ (FDR) < 0.05; Fig. S5*A*]. Further RT-qPCR analysis of more samples revealed that the expression of *COL1A1* and *COL1A2* was not significantly different between the dominant and non-dominant leg ($n = 17$; Fig. S5*B*). We also did not find any significant difference in the percentage of acid-soluble collagen according to which leg the biopsy came from [dominant $2.2 \pm 0.8$ (SEM), non-dominant $1.5 \pm 0.5$ (SEM)] ($n = 17$; Fig. S5*C*).

## Discussion

This is the first time, to our knowledge, that time-dependent expression of circadian clock genes has been demonstrated in human tendon tissues *in vivo*. Further, this is the first evidence of a disrupted circadian rhythm in human chronic tendinopathy, corroborating the idea that a robust circadian rhythm is necessary for tendon health.

Human phasing is not an exact 12 h difference because people are subjected to modern day living entrainment (Hughey & Butte, 2016); however, the expression of core clock genes is an accurate indicator of internal circadian time (Hughey et al., 2016). Our primary goal was to establish if human tendon tissue is a circadian clock. Therefore, our study design was based on only two biopsies from each subject. As predicted, two time points were sufficient for the identification of circadian clock genes to be enriched in the differentially expressed RNAs and the phasing of the clock genes was consistent with that described for human skin, another peripheral tissue clock (Wu et al., 2018). Preferably, we would have liked to perform repeated biopsies at several time points around the clock in the same individual, but this would not have provided reliable results, as the tendon biopsy procedure causes local tissue trauma that persists for several months and alters the gene expression within the tissue (Heinemeier et al., 2016), and so only two biopsies from the same tendon type (one from each limb) can be obtained from any individual.

The reasons why tendinopathy develops are still unclear, but interestingly in the present study we demonstrate that chronic tendinopathy is accompanied by a disrupted circadian rhythm, although it is clearly not possible to elucidate which happened first. Work performed on tendons from mice with mutations in circadian clock genes has demonstrated that a robust circadian rhythm is necessary for tendon homeostasis (Chang et al., 2020; Yeung et al., 2014), which would suggest that dampening of circadian outputs precedes tendinopathy development. On the other hand, it has been demonstrated that endoplasmic reticulum stress

from expression of mutant collagen molecules causes a reduction in circadian clock output in tendon fibroblasts (Pickard et al., 2019), raising the possibility that other stressors that perturb ECM homeostasis and contribute to the development of tendinopathy may also be responsible for a diminished circadian clock. Further, how the tendon clock is entrained is still unknown.

In the present study the contralateral tendon was not clinically considered as chronic tendinopathy, but we observed that it frequently displayed symptoms such as pain and hypervascularisation, unlike tendons in the participating healthy individuals. The loss of night reduction in expression of collagen I fibrillogenesis genes in both the tendinopathy and the contralateral tendon is suggestive of perturbated homeostasis, a phenomenon that could precede the development of tendinopathy. This idea would fit with earlier observations of human patellar tendinopathy where we showed that hyper-vascularisation was already present at the very early stage of the disease (Tran et al., 2020). Clearly, it cannot be ruled out that reduced habitual physical activity as a result of painful chronic tendinopathy may lead to a dampened rest/activity cycle, causing the lack of day–night differences in the expression of collagen I genes. However, the mean activity level of tendinopathy patients was not significantly lower than that of healthy controls. Nonetheless, together with monitoring of circadian clock genes, these moderated collagen I transcript levels could potentially be used as a preclinical biomarker of tendon disease.

The main reason why we could not directly compare the RNAseq analysis of healthy tendons and tendinopathic tendons was because it was not possible to perform pairwise analysis on the chronic tendinopathy samples. Still, the finding of a reduced number of differentially expressed clock genes and lack of diurnal variation in collagen expression in tendon tissue taken from tendinopathic patients might indicate a propensity for individuals with a dampened peripheral circadian rhythm to develop disease. Disturbed homeostasis in tendinopathy may lead to matrix protein excess (e.g. fibrosis) and/or an altered ECM composition (e.g. increased proteoglycans) that may lead to subsequent fluid accumulation and increased tendon size (Malmgaard-Clausen et al., 2021; Tran et al., 2020). We have recently demonstrated that autophagy is vital for normal tendon collagen homeostasis, and that accelerated degradation of collagen and increased autophagy-mediated removal of damaged endoplasmic reticulum (ER-phagy) led to an overall reduction in collagen synthesis (Montagna et al., 2022). This could at least partly explain why no compensatory increased collagen synthesis was observed in tendinopathy in the present study. This suggestion is further supported by the finding, in animal studies, that a healthy ER system is

vital for protecting the circadian rhythm in tendon cells (Pickard et al., 2019). Although the present study cannot fully enlighten the cause–effect relationship between a disturbed circadian clock and development of clinical tendinopathy, our findings indicate that circadian control of tendon is impaired in tendinopathy and that diurnal variation in collagen expression is disturbed.

Compared to a 48-h time-series microarray analysis on mouse tail tendons (Yeung et al., 2014), our current investigation yielded only a moderate number of time-dependent RNAs (~1%) and only ~10% of these overlapped with the mouse dataset. This disparity is likely to be due to analysing only two time points. First, comparing only two time points 12 h apart in a 24 h period would only identify CCGs that are in or close to their peak and nadir expression phases. For example, RT-qPCR analysis, which is far less sensitive than RNAseq, revealed only two of the five genes we measured to be significantly different when the biopsies were analysed together, but by analysing the biopsies separately according to ZT it was possible to demonstrate the limitation of comparing only two time points and the importance of knowing which time points are most appropriate; for example, *CLOCK* expression was not significantly different when comparing ZT1 and ZT13 but was significantly different when comparing ZT2 and ZT14 (Table S2). Second, comparing day and night samples uncovered additional day–night genes that were not under direct regulation of the endogenous circadian clock, that is non-CCGs. For instance, we found expression of *COL1A1*, *COL1A2* and *COL5A1* to be regulated differentially in the day *versus* night but remained independent of the circadian clock. Increased collagen I synthesis in human tendons in response to physical training has been well documented (Heinemeier et al., 2013; Langberg et al., 1999; Miller et al., 2005) and therefore we hypothesise that the reduced collagen I expression in the night biopsies is due to less physical activity in relation to the natural rest–activity cycle.

The observation of differences in gene expression between dominant and non-dominant leg is interesting. The three genes that were found to be upregulated in the dominating leg could align with a more active and explosive use of that leg (e.g. *COL1A2*, which was no longer apparent when more samples were analysed). It is very likely that it was by chance that out of more than 16 000 RNAs there were only nine that showed a difference.

The limitation in sampling repeatedly in the same tendon, the sampling frequency of only two time points (sampling in each of the patellar tendons of the same individual) and the methods used were insufficient for detecting subtle changes of the tendon chronomatrix in human tendons. However, a tendency for an increased number of narrow-diameter fibrils in the night biopsies

was observed. The chronomatrix described for mouse comprises ~3% volume of the tendon ECM; the percentage of narrow-diameter fibrils observed differed from ~30 to ~50% at times of nadir and peak abundance when dispersed fibrils were analysed (Chang et al., 2020), and in the present study the difference between the two time points examined was less and this was probably because the samples were not collected at precisely the peak and nadir phase of chronomatrix production in human patellar tendon; therefore, a longitudinal time series might increase the chances of detecting circadian rhythmicity in collagen turnover.

This knowledge of a circadian rhythm in human tendon tissues and a potential disturbance hereof in tendinopathic tendons have important translational and clinical relevance. Restoration of the circadian rhythm, for example via pharmacological treatment, may prove effective in future treatment of tendon overuse injury. As administration of glucocorticoids in cell cultures ensures synchronisation of circadian oscillators (Balsalobre et al., 2000), the present study speaks in favour of explaining the short-term beneficial effects of glucocorticoids on painful tendinopathy.

In conclusion, diurnal changes in gene expression in healthy human patellar tendons indicate a conserved circadian clock as well as existence of a night reduction in collagen I expression. Fewer time-dependent genes were observed in chronic tendinopathy. A strong direct link between a robust circadian rhythm and tendon health in humans opens up the possibility that pharmacological and non-pharmacological targeting of the tendon circadian clock could be a good preclinical and therapeutic strategy in the future.

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

## Additional information

### Data availability statement

The RNAseq data have been deposited in Array-Express with the data set identifiers E-MTAB-11937 for the healthy tendon and E-MTAB-11938 for the chronic tendinopathy datasets. All data are available in the Supporting information. Data used to generate figures and statistics can be found in Data file S1.

### Competing interests

The authors declare no competing interests.

### Author contributions

C.Y.C.Y. conceived and designed the study and the data analysis, supervised the study, analysed and interpreted the data, and wrote and revised the manuscript. R.B.S. designed data analysis, analysed and interpreted the data, and reviewed the manuscript. K.Y., A.J.U. and Y.L. performed experiments and collected data. I.T., N.M.M.C., M.L. and J.L.O. recruited the participants and collected the biopsies. K.E.K., conceived the fibril measurement study. P.S. designed data analysis, analysed and interpreted the data, and reviewed the manuscript. M.K. conceived and supervised the study, analysed and interpreted the data, and reviewed and revised the manuscript. All authors approved the manuscript.

### Funding

This work was supported by a Lundbeck Foundation Grant awarded to M.K. (R198-2015-207) and the Nordea Foundation (to Centre for Healthy Aging). Y.H. and K.E.K. are supported by Wellcome Trust Investigator and Wellcome Centre Core Awards to K.E.K. (110126/Z/15/Z and 203128/Z/16/Z).

### Acknowledgements

TEM was performed in the Electron Microscopy Facility, Faculty of Biology, Medicine and Health (University of Manchester). The authors would like to thank M. B. Kjær, A. E. M. Jørgensen, C. Couppé and M. Jensen (Institute of Sports Medicine Copenhagen), the Bispebjerg Hospital Laboratory for Stereology and Neuroscience, Bispebjerg Hospital Department of Orthopaedic Surgery, Bispebjerg Hospital Department of Dermatology, Q.-J. Meng (University of Manchester), J. Hogenesch and G. Wu (Cincinnati Children's Hospital Medical Centre) for helpful discussions, providing technical support and/or access to equipment.

### Keywords

chronomatrix, circadian clock, collagen, electron microscopy, human, tendinopathy, tendon, RNAseq

### Supporting information

Additional supporting information can be found online in the Supporting Information section at the end of the HTML view of the article. Supporting information files available:

**Statistical Summary Document**
**Peer Review History**
**Data file S1**
**Data file S2**
**Data file S3**
**Data file S4**
**Data file S5**
**Supporting information**

