## [Peer Review History · The Journal of Physiology]

Disruption of day-to-night changes in circadian gene expression with chronic tendinopathy

Chloé Yeung, Rene B. Svensson, Kateryna Yurchenko, Nikolaj M. Malmgaard-Clausen, Ida Tryggedsson, Marius Lendal, Anja Jokipii-Utzon, Jens Olesen, Yinhui Lu, Karl Kadler, Peter Schjerling, and Michael Kjaer

DOI: 10.1113/JP284083

Corresponding author(s): Chloé Yeung (chloe.yeung@gmail.com)

The following individual(s) involved in review of this submission have agreed to reveal their identity: Mark Viggars (Referee #3)

Review Timeline:

Submission Date:	08-Nov-2022
Editorial Decision:	12-Jan-2023
Revision Received:	26-Jan-2023
Accepted:	08-Feb-2023

Senior Editor: Michael Hogan

Reviewing Editor: Martino Franchi

Transaction Report:

Dear Dr Yeung,

Re: JP-RP-2022-284083 "Disruption of day-to-night changes in circadian gene expression with chronic tendinopathy" by Chloé Yeung, Rene B. Svensson, Kateryna Yurchenko, Nikolaj M. Malmgaard-Clausen, Ida Tryggedsson, Marius Lendal, Anja Jokipii-Utzon, Jens Olesen, Yinhui Lu, Karl Kadler, Peter Schjerling, and Michael Kjaer

Thank you for submitting your manuscript to The Journal of Physiology. It has been assessed by a Reviewing Editor and by 2 expert referees and we are pleased to tell you that it is acceptable for publication following satisfactory revision.

TRANSPARENT PEER REVIEW POLICY: To improve the transparency of its peer review process, The Journal of Physiology publishes online the peer review history of all articles accepted for publication as supporting information. Readers will have access to decision letters, including Editors' comments and referee reports, for each version of the manuscript, as well as any author responses to peer review comments. Referees can decide whether or not they wish to be named on the peer review history document.

REVISION CHECKLIST:

We look forward to receiving your revised submission.

Yours sincerely,

Michael C. Hogan
Senior Editor
The Journal of Physiology
<https://jp.msubmit.net>
<http://jp.physoc.org>
The Physiological Society
Hodgkin Huxley House
30 Farringdon Lane
London, EC1R 3AW
UK
<http://www.physoc.org>
<http://journals.physoc.org>

REQUIRED ITEMS

- Author photo and profile. First (or joint first) authors are asked to provide a short biography (no more than 100 words for one author or 150 words in total for joint first authors) and a portrait photograph. These should be uploaded and clearly labelled with the revised version of the manuscript. See Information for Authors for further details.
- The contact information provided for the person responsible for 'Research Governance' at your institution is an author on this paper. Please provide an alternative contact who is not an author on this paper or confirm that the author whose email was provided has sole responsibility for research governance. This is the person who is responsible for regulations, principles and standards of good practice in research carried out at the institution, for instance the ethical treatment of animals, the keeping of proper experimental records or the reporting of results.
- The Reference List must be in Journal format.
- The Journal of Physiology funds authors of provisionally accepted papers to use the premium BioRender site to create high resolution schematic figures. Follow this link and enter your details and the manuscript number to create and download figures. Upload these as the figure files for your revised submission. If you choose not to take up this offer we require figures to be of similar quality and resolution. If you are opting out of this service to authors, state this in the Comments section on the Detailed Information page of the submission form. The link provided should only be used for the purposes of this submission. Authors will be charged for figures created on this premium BioRender account if they are not related to this manuscript submission.
- Please upload separate high-quality figure files via the submission form.
- Please ensure that any tables are in Word format and are, wherever possible, embedded in the article file itself.
- A Statistical Summary Document, summarising the statistics presented in the manuscript, is required upon revision. It must be on the Journal's template, which can be downloaded from the link in the Statistical Summary Document section here: https://jp.msubmit.net/cgi-bin/main.plex?form_type=display_requirements#statistics.
- Papers must comply with the Statistics Policy: https://jp.msubmit.net/cgi-bin/main.plex?form_type=display_requirements#statistics.

In summary:

- If n {less than or equal to} 30, all data points must be plotted in the figure in a way that reveals their range and distribution. A bar graph with data points overlaid, a box and whisker plot or a violin plot (preferably with data points included) are acceptable formats.
- If $n > 30$, then the entire raw dataset must be made available either as supporting information, or hosted on a not-for-profit repository e.g. FigShare, with access details provided in the manuscript.
- 'n' clearly defined (e.g. x cells from y slices in z animals) in the Methods. Authors should be mindful of pseudoreplication.

- All relevant 'n' values must be clearly stated in the main text, figures and tables, and the Statistical Summary Document (required upon revision).
- The most appropriate summary statistic (e.g. mean or median and standard deviation) must be used. Standard Error of the Mean (SEM) alone is not permitted.
- Exact p values must be stated. Authors must not use 'greater than' or 'less than'. Exact p values must be stated to three significant figures even when 'no statistical significance' is claimed.
- Statistics Summary Document completed appropriately upon revision.
- Please include an Abstract Figure file, as well as the figure legend text within the main article file. The Abstract Figure is a piece of artwork designed to give readers an immediate understanding of the research and should summarise the main conclusions. If possible, the image should be easily 'readable' from left to right or top to bottom. It should show the physiological relevance of the manuscript so readers can assess the importance and content of its findings. Abstract Figures should not merely recapitulate other figures in the manuscript. Please try to keep the diagram as simple as possible and without superfluous information that may distract from the main conclusion(s). Abstract Figures must be provided by authors no later than the revised manuscript stage and should be uploaded as a separate file during online submission labelled as File Type 'Abstract Figure'. Please ensure that you include the figure legend in the main article file. All Abstract Figures should be created using BioRender. Authors should use The Journal's premium BioRender account to export high-resolution images. Details on how to use and access the premium account are included as part of this email.

EDITOR COMMENTS

Reviewing Editor:

The article titled "Disruption of day-to-night changes in circadian gene expression with chronic tendinopathy" provides novel data on the presence of circadian clock within tendon tissue - although, as found by myself and one of the reviewers, this may in part be due to activity patterns. Nevertheless, circadian clock related genes are differentially expressed depending on time of day, and furthermore these same genes are suppressed in those suffering from tendinopathy. Thus, this article possesses a very important translational character and meaning, and I would kindly invite the authors to stress this side too.

As also pointed out by the reviewers, the authors utilise a very thorough methodology throughout the article, investigating their hypotheses in-vivo with human participants and biopsied tendon tissue, and with in-vitro experiments to further investigating into the temporal patterns which is not otherwise possible in-vivo due to biopsy limits.

I therefore ask the authors to respond to the reviewers comments and address the minor points raised.

REFEREE COMMENTS

Referee #2:

I wish to thank the authors for submitting their manuscript entitled "Disruption of day-to-night changes in circadian gene expression with chronic tendinopathy" to the Journal of Physiology. I believe this article is of particular interest to the readers of the Journal. Overall, this manuscript is very well written and provides an appropriately balanced representation of the data. It is fascinating to see that there is a clear circadian clock within tendon tissue - although as discussed by the authors, and by my comments below, this may in part be due to activity patterns. Nonetheless, circadian clock related genes are differentially expressed depending on time of day, and furthermore these same genes are suppressed in those suffering from tendinopathy. The authors utilise a very thorough methodology throughout, predominately investigating their hypotheses in-vivo with human participants and biopsied tendon tissue. However, in-vitro experiments were also utilised in order to further investigating into the temporal patterns which is not otherwise possible in-vivo due to biopsy limits.

I do have several questions and queries in regard to this manuscript particularly where, in my opinion, further detail could be provided. Further, there is one general comment and suggestion for the authors to consider. I hope that the authors find the below comments constructive, as my aim as a reviewer is to aid the manuscript.

General comment:

I strongly appreciate the scientific intrigue behind understanding circadian patterns in tendon tissue, and the differences presented herein with those with tendinopathy. However, I feel that the authors could expand on why specifically this knowledge is important in a clinical setting. I.e., will this new knowledge influence or change treatment pathway of tendinopathy? This addition could come either within the introduction and hence providing further rationale, or alternatively as a small sub-section within the discussion. This is particularly relevant for the potential readers as the authors are

investigating a clinical population.

Specific comments:

Introduction

1. Page 2, Lines 22 "up to 44% in some tissues": could you provide the direct example here for clarity, i.e., which tissue contains 44% circadian clock-controlled genes?
2. Page 2, Lines 34-35: Here you discuss ultrastructural and viscoelastic properties which demonstrate diurnal changes. However, it is not clear whether these are impacted during the day or night, i.e., stiffness is lower at night vs day. Whereas you directly state that a higher abundance of collagen-1 processing occurs in the day. Although minor, it may help the reader to know whether these changes in ultrastructural and viscoelastic properties are lower/higher in the day/night.
3. Page 2, Lines 38: Similar to above, perhaps worth just stating "(reduced stiffness at night)".

Methods:

4. In relation to the Zeitgeber time, I commend this approach as it inherently accounts for the variability in times due to recruitment throughout the year and hence altered circadian patterns. I wondered specifically how these groupings were decided, for example you state there are three groups, ZT1 vs ZT13, ZT2 vs ZT14, and ZT4 vs ZT16, was this simply a case of if ZT = 0-1.49 then ZT1, and 1.5-2.49 is ZT2 etc? Were there any individuals that fell in the middle of two groups and if so, how was this decided?
5. The authors state that cells were cultured for 2 days. Whilst minor, for the sake of replication, a little more detail should be provided here, confluency rates etc.
6. There is little information in regard to how the participants were controlled prior to the biopsies. In the first instance, what measures were taken to minimise loading of the tendon prior to biopsy collection (inherently this could influence expression of collagen related RNA). For example, do the participants arrive at the laboratory a period of time prior to the biopsy and then rested?
7. In the case of the 4am biopsies, do participants sleep at home and then attend the lab, or do they sleep at the university? If the former then how do they travel to the labs, is this via taxi/car or is there any walking/cycling or any form of loading of the tendon prior? In reality, the same query applies for all biopsies.
8. Was there any control on diet prior to or in between the tendon biopsies (i.e., 9pm to 9am, or 4am to 4pm) and/or were these participants in a fasted state during biopsy?
9. Do the authors believe that there was any potential impact of abnormal waking and/or interrupting sleep cycles upon variables discussed herein? I appreciate the RNA sequencing data was obtained from the 9pm biopsies to mitigate this impact significantly. However, were any further measures taken to minimise this potential, particularly for the 4am biopsies?
10. Was there any data collected around habitual physical activity for both healthy and tendinopathic patients? I.e., were healthy more active? As you mentioned later on within the discussion, I do wonder how much of this is true circadian control vs differences in activity, at least in relation to collagen expression. I.E., the day/night gene pattern that is seen may be maintained in healthy due to the increased gradient in activity between day and night compared to the tendinopathic patients who may equally as active/inactive in the day and night due to pain etc - thus reducing the differential between day and night.

Results

11. Figure 2B. I am unsure whether this is a typing error, however the mouse tendon-clock controlled genes are listed as 745, whilst those listed in the orange and red sections add to 746.
12. Page 4, Lines 19-26. I would suggest splitting these sentences up into two, i.e., one sentence for those upregulated in the day and another for those upregulated at night. This will aid the digestion of this data, as it is inherently quite text heavy in nature.
13. Page 6, leg dominance section (Lines 14-23). From reading the supplementary figure, I can see that this data is only completed in the healthy individuals. I would suggest the inclusion of this information with this section. Simply something similar to: "Therefore, we analysed the RNAseq and acid-soluble collagen data as a function of leg dominance in the healthy tendon biopsies."
14. Page 7 Lines 20-22: Is tendinopathy more common in those with disrupted sleep/wake cycles and circadian rhythm such as those who work shifts? If so this would add further evidence to your claims.
15. Page 8, Lines 15-18: What are the proposed mechanisms for this reduction in fibril size / the increase in % of narrow-

diameter fibrils?

16. Table S1: Figure 3 suggests that all 3 ZT groups were analysed for RT-qPCR, but in this table, data are only shown for group 1 and 2. Could I confirm whether the age and time data would be identical to that of Group 3 utilised in collagen content assay, or do they differ? In line with this, were the 3 samples lost at RT-qPCR stage from the ZT4-16 group, thus reducing this to 2?

Referee #3:

Yeung & colleagues present the first known data on circadian rhythm disruption in human patellar tendons diagnosed with tendinopathy, as well as from healthy participants. This study improves the fields basic science understanding of tendon physiology and its transcriptional regulation in humans, as well as providing insight into the role circadian rhythms/disruption may play in the development or accompaniment of tendinopathy. The authors circadian tissue sampling likely represents the absolute limit/highest resolution from human patellar tendons and is likely rather challenging to repeat in larger groups so will be an important landmark for both circadian biologists and MSK research.

I believe Yeung & colleagues present what is largely a robust experimental design with clear presentation and discussion of the results. I have a few minor issues which I believe should be addressed before publication.

- Do you have any evidence that circadian clock expression/phase/amplitude in skeletal muscle myonuclei at or near the myotendinous junction may also be disrupted in tendinopathy? Is the expression of core clock factors affected at both sides of the MTJ? If this analysis is possible from previous samples collected from their research group, I believe it would be extremely beneficial to this manuscript and the field.
- Could the authors check with a statistician/expert, whether or not it is appropriate to combine and compare RNA-seq data when libraries have been prepared using two separate techniques. It is likely that riboRNA depletion and Poly-A selection may select/enrich for slightly different mRNA species. Is there evidence in a principal component analysis that these samples may be drastically different, perhaps affecting the interpretation and number of differentially expressed genes. If so, can some sort of normalisation be performed to account for this. If not, I am happy with current analyses and interpretations.
- While the sample size is small, could the authors investigate any potential relationship/correlation with the expression of core clock factors at ZT10 and ZT22 and the severity of tendinopathy-if at all possible.
- If a lower statistical cut-off is used for the RNA-seq analysis ($Q=0.1$), do the samples from the tendinopathy patients show a drastic increase in oscillating gene transcripts? I am wondering whether the reduction in oscillating gene transcripts may be due to a statistical bias rather than biological.
- Can fold changes be presented for the gene transcripts found to be different between dominant/non-dominant legs.
- Could the authors clarify what is meant by 'under direct regulation of the endogenous clock' on lines 1-7 page 8.
- Given the large effects the circadian clock has on metabolism, is there evidence that metabolic/glycolytic gene transcripts are affected with tendinopathy?
- Can the authors clarify whether or not tendon tissue cultures were performed in serum free media. If not, can the authors confirm that the serum in the media does not influence circadian rhythms post dexamethasone treatment? Perhaps these cells do not survive without serum?
- Can the authors confirm how many PCR cycles were performed for library amplification?
- Can authors include all core clock factors in figure S1?

END OF COMMENTS

Confidential Review

08-Nov-2022

Ching-Yan Chloé Yeung, PhD
Institute of Sports Medicine Copenhagen
Bispebjerg Hospital
Nielsine Nielsens Vej 11
Building 8
Copenhagen NV 2400
Denmark
chloe.yeung@gmail.com

26 January 2023

The Journal of Physiology

Dear Dr Hogan,

RE: Manuscript JP-RP-2022-284083

Please extend our thanks to the Reviewing Editor and the two expert referees for their time in reviewing our manuscript and for their encouraging comments.

Please find attached the revised manuscript, “**Disruption of day-to-night changes in circadian gene expression with chronic tendinopathy**”. The file names “Yeung_tracked_changes” contain changes and additional text highlighted (using Tracked Changes). The cleaned-up version of the manuscript is named “Yeung_revision”. The referencing style has been updated to match *The Journal*.

Please also find the revised versions of figures and files (listed below):

Fig. 2 – type in Fig. 2B corrected

Supplementary Figures PDF – To contain new Fig S4, which shows new data comparing time spent on physical activity by the study participants

Data S1 – Revised to contain data for new Fig. S4

Please also find attached the Author Profile and photo, the Abstract Figure and the Statistical Summary Document.

Below is a detailed response to the reviewers' comments.

Best wishes,

Chloé Yeung

Senior Researcher at Institute of Sports Medicine Copenhagen and Center for Healthy Aging

Manuscript JP-RP-2022-284083 “Disruption of day-to-night changes in circadian gene expression with chronic tendinopathy”

Authors’ responses in blue. Edited/additional text in the manuscript is highlighted here in red.

EDITOR COMMENTS

Reviewing Editor:

The article titled "Disruption of day-to-night changes in circadian gene expression with chronic tendinopathy" provides novel data on the presence of circadian clock within tendon tissue - although, as found by myself and one of the reviewers, this may in part be due to activity patterns. Nevertheless, circadian clock related genes are differentially expressed depending on time of day, and furthermore these same genes are suppressed in those suffering from tendinopathy. Thus, this article possesses a very important translational character and meaning, and I would kindly invite the authors to stress this side too.

As also pointed out by the reviewers, the authors utilise a very thorough methodology throughout the article, investigating their hypotheses in-vivo with human participants and biopsied tendon tissue, and with in-vitro experiments to further investigating into the temporal patterns which is not otherwise possible in-vivo due to biopsy limits.

I therefore ask the authors to respond to the reviewers comments and address the minor points raised.

We thank the Reviewing Editor for their time taken to review our manuscript and for the encouraging comments. We have now added new text to stress the important translational aspect of the study (see response to Reviewer 2's General comment) and addressed the reviewers' comments below.

REFEREE COMMENTS

Referee #2:

I wish to thank the authors for submitting their manuscript entitled "Disruption of day-to-night changes in circadian gene expression with chronic tendinopathy" to the Journal of Physiology. I believe this article is of particular interest to the readers of the Journal. Overall, this manuscript is very well written and provides an appropriately balanced representation of the data. It is fascinating to see that there is a clear circadian clock within tendon tissue - although as discussed by the authors, and by my comments below, this may in part be due to activity patterns. Nonetheless, circadian clock related genes are differentially expressed depending on time of day, and furthermore these same genes are suppressed in those suffering from tendinopathy. The authors utilise a very thorough methodology throughout, predominately investigating their hypotheses in-vivo with human participants and biopsied tendon tissue. However, in-vitro experiments were also utilised in order to further investigating into the temporal patterns which is not otherwise possible in-vivo due to biopsy limits.

I do have several questions and queries in regard to this manuscript particularly where, in my opinion, further detail could be provided. Further, there is one general comment and suggestion for the authors to consider. I hope that the authors find the below comments constructive, as my aim as a reviewer is to aid the manuscript.

We thank the reviewer for their time taken to review our manuscript and for the encouraging comments. We have addressed the comments below.

General comment:

I strongly appreciate the scientific intrigue behind understanding circadian patterns in tendon tissue, and the differences presented herein with those with tendinopathy. However, I feel that the authors could expand on why specifically this knowledge is important in a clinical setting. I.e., will this new knowledge influence or change treatment pathway of tendinopathy? This addition could come either within the introduction and hence providing further rationale, or alternatively as a small sub-section within the discussion. This is particularly relevant for the potential readers as the authors are investigating a clinical population.

We agree with the reviewer and have now added to the text.

In the Introduction:

“Such knowledge will be important in a clinical setting, as efforts to normalise potentially disrupted circadian gene expression, e.g., through pharmacological intervention, could provide new pathways to counteract and treat tendinopathy in humans.” (Page 2, line 41 to Page 3, line 3)

In the Discussion:

“This knowledge of a circadian rhythm in human tendon tissues and a potential disturbance hereof in tendinopathic tendons have important translational and clinical relevance. Restoration of the circadian rhythm, e.g., via pharmacological treatment, may prove effective in future treatment of tendon overuse injury. As administration of glucocorticoids in cell cultures ensures synchronisation of circadian oscillators (PMID: 11009419), the present study speaks in favour of explaining the short term beneficial effects of glucocorticoids on painful tendinopathy.” (Page 8, lines 8-13)

Specific comments:

Introduction

1. Page 2, Lines 22 "up to 44% in some tissues": could you provide the direct example here for clarity, i.e., which tissue contains 44% circadian clock-controlled genes?

We have now edited the text to include the detail:

“Circadian clock-controlled genes (CCGs) are highly tissue-specific, and are up to 44% of the transcriptome in some tissues (visceral fat), with ~12% of these encoding known drug targets (4, 5).” (Page 2, line 21)

2. Page 2, Lines 34-35: Here you discuss ultrastructural and viscoelastic properties which demonstrate diurnal changes. However, it is not clear whether these are impacted during the day or night, i.e., stiffness is lower at night vs day. Whereas you directly state that a higher abundance of collagen-1 processing occurs in the day. Although minor, it may help the reader to know whether these changes in ultrastructural and viscoelastic properties are lower/higher in the day/night.

We have now clarified this in the text:

“The murine tendon ECM exhibits diurnal changes in ultrastructure (observed as fluctuations in the number of small-diameter (~50 nm) collagen fibrils) and viscoelastic properties (higher energy dissipated during cyclic loading and quicker relaxation at night) and a higher abundance of non-covalently bound, processed collagen-I during the day (8, 11).” (Page 2, lines 33-34)

3. Page 2, Lines 38: Similar to above, perhaps worth just stating "(reduced stiffness at night)".

We have now clarified this in the text:

“Together, these ECM changes represent the “chronomatrix” in tendon. Further, there is evidence that human tendon is a peripheral clock tissue; isolated tenocytes in culture have demonstrated endogenous circadian rhythms (12), circadian genes are expressed *in vivo* (13), and there is reduced stiffness in the evening *in vivo* (14-16).” (Page 2, line 38)

Methods:

4. In relation to the Zeitgeber time, I commend this approach as it inherently accounts for the variability in times due to recruitment throughout the year and hence altered circadian patterns. I wondered specifically how these groupings were decided, for example you state there are three groups, ZT1 vs ZT13, ZT2 vs ZT14, and ZT4 vs ZT16, was this simply a case of if ZT = 0-1.49 then ZT1, and 1.5-2.49 is ZT2 etc? Were there any individuals that fell in the middle of two groups and if so, how was this decided?

We thank the reviewer for appreciating this approach. The groupings were indeed done based on similar ZTs. We have now included the range of ZT times in the Methods and Materials and have highlighted one set of biopsies with ZT times that fell between two groups.

“Based on ZT times, the biopsies were divided into three groups: Group 1: ZT1 vs. ZT13 ($n = 7$; range: 0.45-1.00 h and 12.18-12.85 h, respectively), Group 2: ZT2 vs. ZT14 ($n = 5$; range: 1.23-1.87 h and 12.97-13.93 h, respectively) and Group 3: ZT4 vs. ZT16 ($n = 5$; range: 3.75-4.63 h and 15.57-16.47 h, respectively). The

one sample with an evening biopsy ZT of 12.97 h was put in Group 2 and not Group 1 because the ZT of the morning biopsy (1.23 h) was closer to the morning biopsy ZT range of Group 2." (Page 10, lines 10-13)

5. The authors state that cells were cultured for 2 days. Whilst minor, for the sake of replication, a little more detail should be provided here, confluency rates etc.

For replication, we have provided the starting plating density of 1×10^4 cells/cm².

"For the time-series RNA isolation, 1×10^4 cells/cm² were plated onto 35 mm dishes and cultured in complete medium for 2 days. Then, cells were synchronised using 100 nM dexamethasone (Sigma-Aldrich) in complete medium for 24 h, after which the medium was exchanged for complete medium. RNA was isolated every 4 hours from 12 hours to 56 hours after the medium change." (Page 9, line 41 to page 10 line 3)

6. There is little information in regard to how the participants were controlled prior to the biopsies. In the first instance, what measures were taken to minimise loading of the tendon prior to biopsy collection (inherently this could influence expression of collagen related RNA). For example, do the participants arrive at the laboratory a period of time prior to the biopsy and then rested?

In the present study, the biopsy procedure was controlled in that participants were asked to arrive 5-10 mins before but the participants were not controlled in terms of how much/little they should load their tendons. Our primary outcome was the level of circadian clock gene expression at two time points, and we had randomised biopsy order (evening or morning biopsy first) and leg dominance to potentially to minimise any gene expression differences caused by systemic or habitual factors (i.e., caused by the biopsy procedure or loading difference between dominant and non-dominant legs, respectively). We did not expect the expression of collagen genes to be different between day and night because they are not circadian clock-controlled in murine tendons. Therefore, it was a surprise to observe the significantly lowered expression of *COL1A1* and *COL1A2* in the evening in human tendons.

7. In the case of the 4am biopsies, do participants sleep at home and then attend the lab, or do they sleep at the university? If the former then how do they travel to the labs, is this via taxi/car or is there any walking/cycling or any form of loading of the tendon prior? In reality, the same query applies for all biopsies.

Same as above. To avoid any impact of sleep disturbance on the biopsies, we took the 4 AM biopsies second. We have now specified this in the Methods (see response to comment 9 below).

8. Was there any control on diet prior to or in between the tendon biopsies (i.e., 9pm to 9am, or 4am to 4pm) and/or were these participants in a fasted state during biopsy?

While this is an important control for muscle circadian clock entrainment, it is currently not known what internal or external factors entrain the tendon and we did not collect these data. However, as food intake can influence some peripheral clock rhythms, we will control for these in future studies.

9. Do the authors believe that there was any potential impact of abnormal waking and/or interrupting sleep cycles upon variables discussed herein? I appreciate the RNA sequencing data was obtained from the 9pm biopsies to mitigate this impact significantly. However, were any further measures taken to minimise this potential, particularly for the 4am biopsies?

It is still not known how the tendon clock is entrained but it is likely that central clock outputs, including glucocorticoid release, is involved. Therefore, minimising sleep disruption to participant was a major precaution we took in choosing the 9 am and 9 pm as pointed out by the reviewer. This was also why for the 4 pm and 4 am biopsies, we took the sleep-disrupting 4 am biopsies as the second biopsy to avoid any downstream effect on the day biopsies. No other measures were taken to minimise impact on the samples because the circadian clock system is designed so that the circadian rhythm of peripheral clocks can remain robust to allow for noise in the environment (i.e., sudden changes to external entrainment signals), preventing unwanted phase shifts. This robustness is also why human individuals are slow to adapt to changes such as jet lag. For an external time cues or entrainment factors to impact on a peripheral circadian rhythm (i.e., act as a zeitgeber), they have to be sustained changes.

"For the TEM, the first leg (dominant or non-dominant) to be biopsied was alternated between participants at 4 PM. The second biopsy was taken from the other leg at 4 AM. The 4 AM biopsies were taken as second biopsies to avoid the possibility of a disrupted sleep or circadian rhythm impacting on the next biopsy." (Page 9, lines 27-28)

10. Was there any data collected around habitual physical activity for both healthy and tendinopathic patients? I.e., were healthy more active? As you mentioned later on within the discussion, I do wonder how much of this is true circadian control vs differences in activity, at least in relation to collagen expression. I.E., the day/night gene pattern that is seen may be maintained in healthy due to the increased gradient in activity between day and night compared to the tendinopathic patients who may equally as active/inactive in the day and night due to pain etc - thus reducing the differential between day and night.

We thank the reviewer for this important point. We had collected data regarding the number of hours spent on physical activity per week from all participants. We found no significant difference when we compared the how long healthy and tendinopathic participants were physically active around the time when the biopsies were sampled. However, when we compared tendinopathic patients' physical activity times before and after they became affected with tendinopathy we found that they had significantly reduced time spent on being physically active, indicating a reduced habitual activity. We have now incorporated this data as a new Fig. S4 and in the main text.

In the Methods:

"All participants were asked to how many hours per week they spend on physical activity and patients with chronic tendinopathy were additionally asked how many hours they used to spend on physical activity." (Page 9, lines 16-18)

In the Results:

"To address whether reduced activity level in chronic tendinopathy patients compared to healthy participants was the reason why no time-dependent significant differences were found in the expression of collagen genes, we compared the amount of time they spent on physical activity and found no significant difference ($n = 17$ for healthy and $n = 10$ for chronic tendinopathy; Fig. S4; 7.6 ± 3.9 (SD) h per week in healthy, 6.1 ± 4.2 (SD) h per week in tendinopathy). However, we found that participants with tendinopathy had significantly reduced their habitual physical activity when they began to develop symptoms ($n = 10$; Fig. S4; 9.5 ± 7.8 (SD) h per week before tendinopathy)." (Page 5, lines 27-33)

In the Discussion:

"Clearly, it cannot be ruled out that reduced habitual physical activity as a result of painful chronic tendinopathy may lead to a dampened rest/activity cycle, causing the lack of day-night difference in the expression of collagen I genes. However, the mean activity level of tendinopathy patients was not significantly lower than that of healthy controls." (Page 6, lines 36-39)

Results

11. Figure 2B. I am unsure whether this is a typing error, however the mouse tendon-clock controlled genes are listed as 745, whilst those listed in the orange and red sections add to 746.

The reviewer is correct, and we have now rectified this typo. Please see new Fig. 2.

12. Page 4, Lines 19-26. I would suggest splitting these sentences up into two, i.e., one sentence for those upregulated in the day and another for those upregulated at night. This will aid the digestion of this data, as it is inherently quite text heavy in nature.

We have now changed this as suggested:

"In total we found 11 conserved circadian genes. The expression of *BHLHE41* (basic helix-loop-helix family member E41), *CIART* (circadian associated repressor of transcription), *DBP* (D-box binding PAR BZIP transcription factor), *NR1D1* (nuclear receptor subfamily 1 group D member 1), *NR1D2* (nuclear receptor subfamily 1 group D member 2), *PER3* (period 3), and *TEF* (thyrotroph embryonic factor) was upregulated in the day (Fig. 2C). And the expression of *ARNTL* (aryl hydrocarbon receptor nuclear translocator like-1, also known as brain and muscle ARNT-like 1 (BMAL1)), *CRY1* (cryptochrome circadian regulator 1), *GHRL* (ghrelin and obestatin prepropeptide) and *NFIL3* (nuclear factor, interleukin 3 regulated) were upregulated at night (Fig. 2D)." (Page 3 lines 32-39)

13. Page 6, leg dominance section (Lines 14-23). From reading the supplementary figure, I can see that this data is only completed in the healthy individuals. I would suggest the inclusion of this information with this section. Simply something similar to: "Therefore, we analysed the RNAseq and acid-soluble collagen data as a function of leg dominance in the healthy tendon biopsies."

This has now been changed, as suggested.

14. Page 7 Lines 20-22: Is tendinopathy more common in those with disrupted sleep/wake cycles and circadian rhythm such as those who work shifts? If so this would add further evidence to your claims.

This is a great question, and the answer is no one knows. It is also difficult to extrapolate because with shift work the nature of the work tends to be manual, and there are increased number of accidents relating to fatigue, among other factors that could increase the risk of developing tendinopathy.

15. Page 8, Lines 15-18: What are the proposed mechanisms for this reduction in fibril size / the increase in % of narrow-diameter fibrils?

The reviewer refers to the findings of a previous study performed with mouse tendons (PMID: 31907414). In that study, we discussed that the narrow diameter of these collagen I-containing fibrils is likely regulated by other components of the chromatrix that regulate fibril diameter and candidate molecules are small leucine-rich proteoglycans, including decorin, and collagen V, which regulates collagen I fibrillogenesis. In that same study, we proposed that the increase in number/percentage of the narrow-diameter collagen fibrils during the day is due to circadian regulation of collagen-I secretion.

16. Table S1: Figure 3 suggests that all 3 ZT groups were analysed for RT-qPCR, but in this table, data are only shown for group 1 and 2. Could I confirm whether the age and time data would be identical to that of Group 3 utilised in collagen content assay, or do they differ? In line with this, were the 3 samples lost at RT-qPCR stage from the ZT4-16 group, thus reducing this to 2?

All samples with remaining RNA were analysed by RT-qPCR. The data in Tables S1 are only shown for Groups 1 and 2 because there was only $n = 2$ in Group 3 (due to insufficient amount of leftover RNA after RNAseq analysis), which is not enough data points for statistical analysis and thus Group 3 samples were not analysed on their own but they were included in the analysis of all samples (Fig. 3). For the two Group 2 samples analysed by RT-qPCR, the mean age was 20.5 ± 0.7 (SD) years, the wall time of the day biopsies was $09:07 \pm 00:18$ (hh:mm), the wall time of the night biopsies was $21:02 \pm 00:04$ (hh:mm), ZT of day biopsies was 4.4 ± 0.2 h, and ZT of night biopsies was 16.3 ± 0.2 h.

Referee #3:

Yeung & colleagues present the first known data on circadian rhythm disruption in human patellar tendons diagnosed with tendinopathy, as well as from healthy participants. This study improves the fields basic science understanding of tendon physiology and its transcriptional regulation in humans, as well as providing insight into the role circadian rhythms/disruption may play in the development or accompaniment of tendinopathy. The authors circadian tissue sampling likely represents the absolute limit/highest resolution from human patellar tendons and is likely rather challenging to repeat in larger groups so will be an important landmark for both circadian biologists and MSK research.

I believe Yeung & colleagues present what is largely a robust experimental design with clear presentation and discussion of the results. I have a few minor issues which I believe should be addressed before publication.

We thank the reviewer for their time taken to review our manuscript and for the encouraging comments. We have addressed the specific comments below.

- Do you have any evidence that circadian clock expression/phase/amplitude in skeletal muscle myonuclei at or near the myotendinous junction may also be disrupted in tendinopathy? Is the expression of core clock factors affected at both sides of the MTJ? If this analysis is possible from previous samples collected from their research group, I believe it would be extremely beneficial to this manuscript and the field.

This is a great suggestion for a future study looking into the crosstalk of circadian rhythms between neighbouring musculoskeletal tissues. In the current study, only biopsies from the tendon proper were sampled.

- Could the authors check with a statistician/expert, whether or not it is appropriate to combine and compare RNA-seq data when libraries have been prepared using two separate techniques. It is likely that riboRNA depletion and Poly-A selection may select/enrich for slightly different mRNA species. Is there evidence in a

principal component analysis that these samples may be drastically different, perhaps affecting the interpretation and number of differentially expressed genes. If so, can some sort of normalisation be performed to account for this. If not, I am happy with current analyses and interpretations.

We have a statistician expert as a co-author and we agree that it is not optimal to use two different techniques due to a communication error for the first samples. Indeed it makes a large difference for which genes are measured. For instance, the histone genes are high in the riboRNA depletion samples and very low in the poly-A samples due to the lack of a poly-A tail. However, in general the mRNAs are measured at a comparable level (within a few fold) and since the comparisons are always paired (same subject, same method) the effect should be minor as the “method” effect is included in the subject effect and therefore accounted for in the statistical analysis.

- While the sample size is small, could the authors investigate any potential relationship/correlation with the expression of core clock factors at ZT10 and ZT22 and the severity of tendinopathy-if at all possible.

The reviewer raises an interesting and clinically relevant point, and we would love to be able to make such a correlation. However, there is no consensus on what biomarkers, whether these are levels of molecules or gene expression or area of increased doppler flow correlate with the severity of tendinopathy that we may use.

- If a lower statistical cut-off is used for the RNA-seq analysis (Q=0.1), do the samples from the tendinopathy patients show a drastic increase in oscillating gene transcripts? I am wondering whether the reduction in oscillating gene transcripts may be due to a statistical bias rather than biological.

We have looked at the data again and we do not see a drastic increase in oscillating gene transcripts when we lower the statistical cut-off use for the RNAseq analysis (please see plot below).

- Can fold changes be presented for the gene transcripts found to be different between dominant/non-dominant legs.

Thank you for discovering our mistake in Figure S4A. In the legend *P* values were missing for *CD52*, *IGLC2* and *JCHAIN*. And we should of course have clarified that the log2foldchanges (shrunk) were very low for *IGLC2*, *JCHAIN* and *NCMAP* ($-1e^{-5}$) and therefore although significant, not clearly changed.

“(A) Normalised RNA counts for *COL1A2*, *PTGIS* and *SLIT3* were significantly upregulated (log₂ fold change 0.6 to 1.0) in tendon biopsies of the dominant leg ($n = 5$ biological samples; P (FDR) = 0.0447, 0.0062, 0.0447, respectively; from DESeq2 analysis). RNA counts for *CD52*, *IGLC2*, *JCHAIN*, *MPZ*, *NCMAP* and *TREM1* were significantly upregulated (log₂ fold change 1.1 to 1.4) in tendon biopsies of the non-dominant leg ($n = 5$ biological samples; P (FDR) = 0.0062, 0.0062, 0.0062, 0.0447, 0.0447, 0.0078, respectively; from DESeq2 analysis). Although significant, *IGLC2*, *JCHAIN* and *NCMAP* had log₂ fold change of $-1e^{-5}$ (shrunk values) making a real change questionable. Connecting lines indicate paired samples.”

- Could the authors clarify what is meant by 'under direct regulation of the endogenous clock' on lines 1-7

“Second, comparing day and night samples uncovered additional day-night genes that were not under direct regulation of the endogenous circadian clock, i.e., non-CCGs.” (Page 7, line 29)

- Given the large effects the circadian clock has on metabolism, is there evidence that metabolic/glycolytic gene transcripts are affected with tendinopathy?

As we are unable to compare the RNAseq data from healthy with the chronic tendinopathy, it would be very difficult to confidently determine transcripts changed with disease. As stated in the manuscript, the RNAs determined to be time-dependent only showed enrichment for circadian rhythm and transcription and we were unable to determine genes regulating metabolism as significantly enriched.

- Can the authors clarify whether or not tendon tissue cultures were performed in serum free media. If not, can the authors confirm that the serum in the media does not influence circadian rhythms post dexamethasone treatment? Perhaps these cells do not survive without serum?

As stated in the Methods section, the experiments were performed in complete medium, which contains 10% foetal calf serum, ensuring that cells were happy.

- Can the authors confirm how many PCR cycles were performed for library amplification?

The library amplification was performed by Genewiz. The information they can provide us with is that NEBNext Ultra II Directional RNA Library Prep Kit was used to according to manufacturer's protocol https://international.neb.com/-/media/nebus/files/manuals/manuale7760_e7765.pdf?rev=5e2b516c5a2547e6a1345c148bf7432f&hash=5078CA97DC56BE0F5369F8B46DE38DF6 (page 10-11)

The protocol recommends 8-16 cycles according to the RNA input. The total RNA content of our samples were between 200-200 ng, so it could be anywhere between 12-13 cycles recommended for 100 ng of RNA.

- Can authors include all core clock factors in figure S1?

We could perform more PCR for more core clock genes, however, because we are still limited by only having two time points and biological replicates (in Group 3), the information gained will not be meaningful.

Dear Dr Yeung,

Re: JP-RP-2023-284083R1 "Disruption of day-to-night changes in circadian gene expression with chronic tendinopathy" by Chloé Yeung, Rene B. Svensson, Kateryna Yurchenko, Nikolaj M. Malmgaard-Clausen, Ida Tryggedsson, Marius Lendal, Anja Jokipii-Utzon, Jens Olesen, Yinhui Lu, Karl Kadler, Peter Schjerling, and Michael Kjaer

We are pleased to tell you that your paper has been accepted for publication in The Journal of Physiology.

Authors should note that it is too late at this point to offer corrections prior to proofing. The accepted version will be published online, ahead of the copy edited and typeset version being made available. Major corrections at proof stage, such as changes to figures, will be referred to the Editors for approval before they can be incorporated. Only minor changes, such as to style and consistency, should be made at proof stage. Changes that need to be made after proof stage will usually require a formal correction notice.

Yours sincerely,

Michael C. Hogan
Senior Editor
The Journal of Physiology
<https://jp.msubmit.net>
<http://jp.physoc.org>
The Physiological Society
Hodgkin Huxley House
30 Farringdon Lane
London, EC1R 3AW
UK
<http://www.physoc.org>
<http://journals.physoc.org>

P.S. - You can help your research get the attention it deserves! Check out Wiley's free Promotion Guide for best-practice recommendations for promoting your work at www.wileyauthors.com/eeo/guide. You can learn more about Wiley Editing Services which offers professional video, design, and writing services to create shareable video abstracts, infographics, conference posters, lay summaries, and research news stories for your research at www.wileyauthors.com/eeo/promotion.

IMPORTANT NOTICE ABOUT OPEN ACCESS: To assist authors whose funding agencies mandate public access to published research findings sooner than 12 months after publication, The Journal of Physiology allows authors to pay an Open Access (OA) fee to have their papers made freely available immediately on publication.

You can check if your funder or institution has a Wiley Open Access Account here: <https://authorservices.wiley.com/author-resources/Journal-Authors/licensing-and-open-access/open-access/author-compliance-tool.html>.

EDITOR COMMENTS

Reviewing Editor:

The authors have addressed comments made by the reviewers in a satisfactory way. I have nothing more to add.

REFEREE COMMENTS

Referee #2:

Dear Authors,

I thank you for the thorough rebuttal. I am happy with the changes made within the manuscript and appreciate the detailed discussion/rationale provided in response to my queries and comments. I have nothing further to add, other than my congratulations for a very nice paper.

Referee #3:

I would like to thank the authors for promptly amending the manuscript, I have no further concerns to address. This is a fantastic body of work and I look forward to seeing more on this topic from this group.

1st Confidential Review

26-Jan-2023